# Single-cell transcriptomic atlas reveals increased regeneration in diseased human inner ear balance organs

Tian Wang[1,2], Angela H. Ling[1,3], Sara E. Billings [1], Davood K. Hosseini [1], Yona Vaisbuch[1], Grace S. Kim[1], Patrick J. Atkinson[1], Zahra N. Sayyid [1], Ksenia A. Aaron[1], Dhananjay Wagh[4], Nicole Pham [1], Mirko Scheibinger[1], Ruiqi Zhou[3], Akira Ishiyama[5], Lindsay S. Moore[1], Peter Santa Maria[1], Nikolas H. Blevins[1], Robert K. Jackler[1], Jennifer C. Alyono[1], John Kveton[6], Dhasakumar Navaratnam [6,7], Stefan Heller [1], Ivan A. Lopez [5], Nicolas Grillet [1], Taha A. Jan [3] ✉ & Alan G. Cheng [1]✉

Mammalian inner ear hair cell loss leads to permanent hearing and balance dysfunction. In contrast to the cochlea, vestibular hair cells of the murine utricle have some regenerative capacity. Whether human utricular hair cells regenerate in vivo remains unknown. Here we procured live, mature utricles from organ donors and vestibular schwannoma patients, and present a validated single-cell transcriptomic atlas at unprecedented resolution. We describe markers of 13 sensory and non-sensory cell types, with partial overlap and correlation between transcriptomes of human and mouse hair cells and supporting cells. We further uncover transcriptomes unique to hair cell precursors, which are unexpectedly 14-fold more abundant in vestibular schwannoma utricles, demonstrating the existence of ongoing regeneration in humans. Lastly, supporting cell-to-hair cell trajectory analysis revealed 5 distinct patterns of dynamic gene expression and associated pathways, including Wnt and IGF-1 signaling. Our dataset constitutes a foundational resource, accessible via a web-based interface, serving to advance knowledge of the normal and diseased human inner ear.

Hearing and vestibular disorders affect over 460 million people worldwide[1]. In the inner ear, auditory and vestibular hair cells are essential for detecting sound and head motions. Genetic mutations, ototoxins, noise exposure, and aging are known to cause hair cell degeneration, leading to permanent hearing loss and vestibular dysfunction[2–4]. While the loss of mammalian auditory hair cells is irreversible, mouse models have shown that vestibular hair cells turn over during homeostasis and regenerate after damage, albeit to a limited degree[5–12]. Prior studies examining the regenerative capacity of the human inner ear have been limited to showing hair cell regeneration using aminoglycoside-damaged organotypic cultures of utricles procured from surgical patients[13,14]. Whether these phenomena

[1]Department of Otolaryngology – Head and Neck Surgery, Stanford University School of Medicine, Stanford, CA 94305, USA. [2]Department of Otolaryngology – Head and Neck Surgery, The Second Xiangya Hospital, Central South University, Changsha, Hunan Province, 410011, PR China. [3]Department of Otolaryngology – Head and Neck Surgery, Epithelial Biology Center, Vanderbilt University Medical Center, Nashville, TN 37232, USA. [4]Stanford Genomics Facility, Stanford University School of Medicine, Stanford, CA 94305, USA. [5]Department of Head and Neck Surgery, University of California Los Angeles, Los Angeles, CA 90095, USA. [6]Department of Surgery, Yale University School of Medicine, New Haven, CT 06510, USA. [7]Department of Neurology, Yale University School of Medicine, New Haven, CT 06510, USA. ✉e-mail: taha.a.jan@vumc.org; aglcheng@stanford.edu

similarly occur in the human utricle in vivo has not yet been demonstrated, in part because obtaining normal and diseased human inner ear tissues is exceedingly challenging.

The human inner ear is housed in the temporal bone, making it accessible only via surgery[15,16]. Because biopsy of the human inner ear may render it non-functional, the majority of work done on the human inner ear has been performed on postmortem, cadaveric tissues[16–25]. A few studies on viable inner ear tissues from fetuses and organoids derived from embryonic or induced pluripotent stem cells have characterized morphological and molecular features of the developing human inner ear[26–32], however, the transcriptomic signature of the healthy, adult human inner ear has remained elusive because of the difficulty of obtaining live samples for such experiments.

An alternative approach is to analyze live inner ear tissues extracted during surgery from patients affected by vestibular schwannoma, a benign tumor of the myelin-forming cells of the vestibulocochlear nerve. Patients with vestibular schwannoma develop hearing loss and vestibular dysfunction[33,34], presumably due to spiral ganglion cell loss caused by nerve compression[35] or hair cell degeneration caused by secreted factors[36]. In certain cases, vestibular schwannoma patients warrant surgical resection via a translabyrinthine approach where inner ear tissues are sacrificed[15,37]. Utricles harvested from vestibular schwannoma patients during surgery have served as the primary model system to characterize hair cell morphology[17,23,38], their capacity to regenerate after aminoglycoside damage in vitro[13,14], and responsiveness to Atoh1 overexpression to induce ectopic hair cells[13]. However, it remains undetermined in these diseased samples whether hair cell regeneration occurs spontaneously in human utricles in vivo.

In this study, we successfully characterized and validated utricular single-cell transcriptomes in adult organ donors and vestibular schwannoma patients. Our computational analysis showed that adult human utricles are composed of 13 cell types with unique molecular signatures, with transcriptomes of human hair cells and supporting cells modestly correlating with those in mice. Single-cell transcriptomic trajectory analysis revealed dynamic changes as supporting cells transition into hair cells in both vestibular schwannoma and organ donor utricles. Relative to organ donor utricles, vestibular schwannoma utricles show significantly more hair cell loss and a ~14-fold increase of hair cell precursors (POU4F3+ and GFI1+, MYO7A-negative). Together, we have demonstrated spontaneous, unmanipulated hair cell regeneration in the adult human utricle and the underlying transcriptional changes during homeostasis and in response to damage. These rare datasets represent foundational tools to guide the investigation of human hair cell regeneration and define molecular similarities and differences between human and mouse vestibular organs.

## Results

### Transcriptional heterogeneity of the mature human utricle

To characterize the human utricle, we harvested tissues from two cohorts: organ donors and vestibular schwannoma patients (Fig. 1a, Supplementary Data 1). We previously devised a surgical approach to procure unilateral or bilateral utricles from organ donors (OD)[39,40], who lacked any history of auditory or vestibular dysfunction (Supplementary Data 1). In parallel, utricles were procured from patients with unilateral vestibular schwannomas and ipsilateral auditory and/or vestibular deficits, undergoing a translabyrinthine approach for tumor resection (Fig. 1a and Supplementary Fig. 1a).

Because vestibular schwannoma patients suffer from auditory and/or vestibular deficits and because hair cell degeneration is detected in their cochlear and utricular organs postmortem[17,23], we hypothesized that vestibular schwannoma utricles are damaged whereas those from organ donors may serve as normal controls (Fig. 1a, Supplementary Fig. 1a, Supplementary Data 1). Four vestibular schwannoma and two organ donor utricles were subject to single-cell

isolation and sequencing. Sensory epithelia were enzymatically and mechanically purified from surrounding stromal and neural tissues (Supplementary Fig. 1b), yielding a total of 27,631 single cells that were sequenced (15,309 organ donor cells and 12,322 vestibular schwannoma cells, Fig. 1b', Supplementary Fig. 2a). Samples from eight captures (four vestibular schwannoma samples with one capture each, and two organ donor samples in four captures) were sequenced (10x Genomics platform), and cells underwent quality control metrics and downstream analyses (Supplementary Fig. 1c). Our integrated dataset (both organ donor and vestibular schwannoma) contained a median of 953 genes per cell (Supplementary Fig. 1d) and unsupervised clustering identified 13 cell states (Fig. 1b). We annotated the cell clusters based on known marker genes in the mammalian inner ear (Fig. 1b, Supplementary Fig. 1e), including 13 different cell types with 6645 cluster-defining genes (Wilcoxon test, FDR < 0.01, Fig. 1c): supporting cells (FBXO2, GFAP, KRT19, clusters 1, 5, 6), hair cells (CIB3, GPX2, LRRC10B, ESPN, cluster 7), roof cells (CITED1, cluster 9), dark cells (HBD, KCNQ1, IGF-1, cluster 2), stromal cells (COCH, cluster 0), melanocytes (PMEL, DCT, cluster 11), vascular cells (FLT1, cluster 10), macrophages (CCL3, cluster 3), pericytes (RGS5, VWF, cluster 8), Schwann cells (MBP, cluster 12), and immune cells (SERPINE2, cluster 4).

To further restrict our analyses to cells in the sensory epithelium, we focused on clusters defined as hair cells, putative hair cell precursors, and supporting cells (Supplementary Fig. 2a, a'). As expected, EPCAM expression was robust in these sensory epithelial cell clusters (Fig. 1d). Expression of SOX2, a known marker of supporting cells and type II hair cells in human, was enriched in a subset of hair cells and supporting cells, while expression of MYO7A, a known human generic hair cell marker[38], is the highest in cluster 7 (Fig. 1d). Subsetting and visualization with similarity weighted non-negative embedding (SWNE) showed eight cell states representing utricle sensory epithelial cells (Supplementary Fig. 2a'). SWNE was used for all 2-dimensional visualization of reduced dimensions to optimally capture global and local relationships among cells and clusters[41]. The final analysis consisted of 5837 organ donor and 4358 vestibular schwannoma utricular cells (Supplementary Fig. 2a', c). Differential gene expression analysis identified 5321 cluster-defining genes (Supplementary Fig. 2a'), yielding five subsets of supporting cells (n = 9200 cells, clusters 0, 1, 2, 3, and 4), one putative hair cell precursor group (n = 154 cells, cluster 7), and two hair cell subtypes (n = 498 type I hair cells, cluster 5 and n = 343 type II hair cells, cluster 6) (Supplementary Fig. 2d, e). A bootstrapping algorithm was utilized to generate all cell clustering parameters and determine cluster robustness (Supplementary Fig. 2b-b")[42]. A dot plot illustrating differential gene expression among these 8 clusters is shown in Supplementary Fig. 2f. Together, this dataset represents the molecular identities of distinct cell types of the adult human utricle.

### Molecular identity of human hair cells and supporting cells

The sensory epithelium is comprised of sensory hair cells and non-sensory supporting cells (Fig. 2a). To identify human hair cell- and supporting cell-defining genes, we performed a direct comparison without the putative hair cell precursors (Fig. 2b). We identified 1251 enriched genes in hair cells and 1259 enriched genes in supporting cells with at least 2-fold expression difference (LFC > 2, FDR < 0.01, Fig. 2c). Among the 24 most differentially expressed genes in hair cells (CABP2, CIB3, GPX2, ESPN, CD164L2, SYT14, SNCG, CPE, STRC, SMPX, ABCA5, POU4F3) and supporting cells (CLU, GFAP, TMSB4X, CLDN4, TMSB10, CD9, VIM, ELF3, IFITM3, S100A10, KRT8, ANXA2, Fig. 2d, e), we validated 13 genes by performing in situ hybridization and immunolabeling on utricles procured from other organ donors and vestibular schwannoma patients. Both anti-MYO7A and SYT14 labeled all hair cells in organ donor and vestibular schwannoma samples (Fig. 2f, g, Supplementary Fig. 2g), where fewer hair cells and a flatter epithelium were observed. As in mice, human supporting cell nuclei reside in the basal

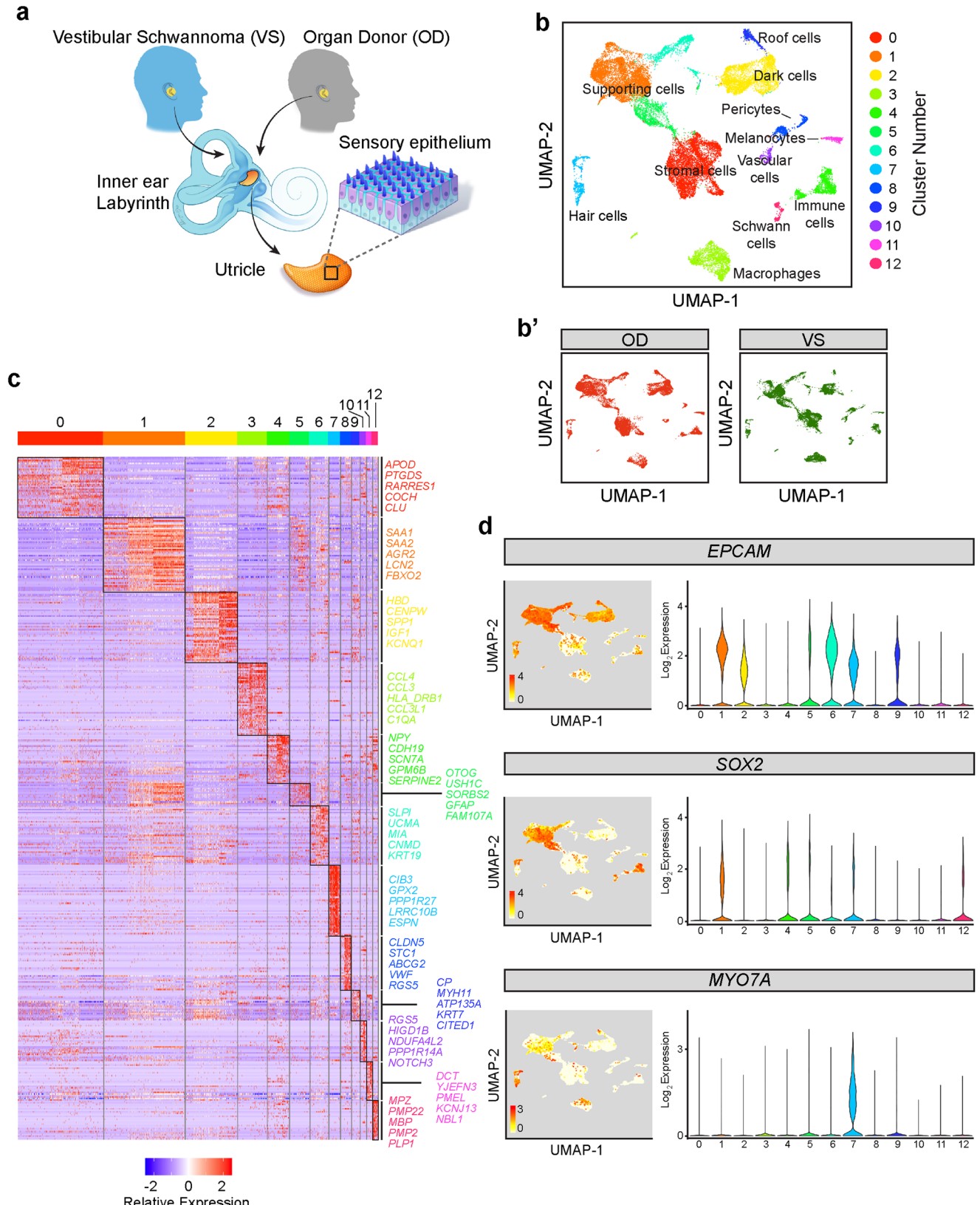

layer of the sensory epithelium and project cytoplasmic processes to the luminal surface (Fig. 2f)[43]. As computationally predicted, both anti-GFAP and *ANXA2* selectively marked supporting cells from organ donor and vestibular schwannoma tissues (Fig. 2f, g). Expression plots confirmed *ANXA2* and *SYT14* as pan-markers of supporting cells and hair cells, respectively (Fig. 2h). Similarly, in situ hybridization

validated *PPP1R27* and *LRRC10B* to be expressed in all hair cells and *CD9* in supporting cells (Supplementary Fig. 2g). Finally, we separately analyzed single cells from organ donor and vestibular schwannoma patients and similarly found individual cell clusters, including hair cells and supporting cells among subsetted sensory epithelial cells (Supplementary Fig. 3a–h). These results highlight common gene

**Fig. 1 | Sensory and non-sensory cell types in utricles from vestibular schwannoma and organ donor patients. a** Cartoon depicting procurement of human utricles from patients undergoing translabyrinthine resection of vestibular schwannoma (VS) and organ donors (OD). Samples from both sources were used for histological and single-cell RNA sequencing analyses. **b** UMAP plot of integrated dataset of four VS utricles and two OD utricles showing 27,631 single cells following all quality control steps including exclusion of doublets. Thirteen cell clusters were identified, and marker genes were used to annotate the sensory and non-sensory cell types. **b'** The UMAP plot from (**b**) is decomposed into the cells originating from OD and VS subjects. There were 15,309 and 12,322 cells from the OD and VS samples, respectively. **c** Heatmap showing the top 50 differentially expressed genes

among the 13 cell clusters. The top differentially expressed genes of each cluster are shown on the right side of the heatmap. A full list of these genes is found in the Source Data file. The heatmap is colored by relative expression from -2 (blue), 0 (white), and 2 (red). **d** Expression of established markers of epithelial cells (*EPCAM*), non-sensory cells and type II hair cells (*SOX2*), and sensory cells (*MYO7A*) within the cell clusters. UMAP plots colored by log$_2$ expression (values of 0 in white to maximum in red as indicated) and violin plots colored by cell cluster are shown. *EPCAM* was highly expressed in clusters 1, 2, 5, 6, 7, and 9; *SOX2* in clusters 1, 4, 5, 6, 7, and 12; and *MYO7A* in clusters 7, which represents hair cells. *SOX2* labels a subgroup of the *MYO7A*⁺ cells indicating type II hair cells. Source data are provided as a Source Data file.

expression profiles of hair cells and supporting cells between human organ donor and vestibular schwannoma utricles through integrated and separate analyses.

## Spatial and transcriptomic properties of hair cell subtypes

Utricular hair cells of both mouse and human fall into two main subtypes (-50% each of type I and II)[44] (Fig. 3a). In mice, this distinction is based on morphology, innervation, composition of ion channels, and gene expression[44–47]. However, in humans, this configuration has only been shown using morphological assessments of postmortem tissues[17,18]. In our integrated dataset of the sensory epithelium, differential gene expression revealed two putative hair cell subpopulations (Supplementary Fig. 2d), which we presumed to represent type I and type II hair cells (Fig. 3b, c). Forty-eight genes were highly expressed in human type I hair cells and 79 genes in type II hair cells (Fig. 3c, LFC > 2 and FDR < 0.01). In addition, we separately analyzed single cells from organ donor and vestibular schwannoma patients and found similar differentially expressed genes among type I and type II hair cells (Supplementary Fig. 3e, j).

We next validated 6 of these differentially expressed genes in the utricular sensory epithelia of organ donors and vestibular schwannoma patients (Fig. 3f–i, Supplementary Fig. 4a, b). In the mouse and human utricular sensory epithelium, hair cell subtypes have distinct shapes: type I hair cells are amphora-shaped and harbor calyx innervation, while type II hair cells are goblet-shaped and display basolateral cytoplasmic processes. Both hair cell subtypes have their nuclei closer to the epithelial surface, contrasting with those of supporting cells that are closer to the basal surface[12,17,22,43] (Fig. 3a).

In organ donor tissues, the morphology of MYO7A⁺ hair cells was maintained and expression of *ADAM11* was exclusive to amphora-shaped type I hair cells with Tuj1⁺ calyceal innervation (Fig. 3f, f'). Type II hair cells displayed basolateral processes and were specifically marked by both anti-SOX2 and *KCNH6* (Fig. 3h, h'). In vestibular schwannoma patients, despite the loss of lamination and cell loss, both *ADAM11* and *KCNH6* expression are similarly restricted to type I (Tuj1⁺ calyceal innervated) and II hair cells (SOX2⁺), respectively (Fig. 3g, g', i, i'). These results validate the transcriptomic data showing enrichment of *ADAM11* and *KCNH6* in type I and II hair cells, respectively, in both cohorts (Fig. 3j). Furthermore, we validated the expression of enriched type I hair cell (*CRABP1* and *VSIG10L2*) and type II hair cell genes (*CALB1*) (Supplementary Fig. 4a, b). These results confirmed the presence of two transcriptionally distinct human utricular hair cell subtypes in both organ donor and vestibular schwannoma utricles.

## Comparison of mouse and human utricular cell transcriptomes

To assess the similarities among the transcriptomes of human and mouse utricular hair cells and supporting cells, we compared our integrated dataset to that recently generated from the adult mouse utricle[48]. We performed a rank Spearman correlation analysis using the intersection of differentially expressed genes between human and mouse cell types[49]. Among genes expressed in human hair cells subtypes, only a minority is shared with those in mouse hair cell subtypes (25.9 and 32.3% of type I and II hair cells, respectively) (Supplementary

Fig. 4c–e). Among the 13 most enriched type I hair cell genes in human utricle (*VSIG10L2, BMP2, CLDN5, ALDH1A3, TAC1, FBXW7, CRABP1, OCM, ATP2B2, PGA5, LINC02568, PRKCD, ADAM11*, Fig. 3d), nine have been reported in the mouse inner ear but only three showed expression restricted to type I hair cells[45–48].

Reciprocally, we examined genes in mouse utricular hair cell subtypes and compared them to our human dataset, and found that some type I hair cell genes (45.7%) and only 29.8% of type II hair cell genes are shared (Supplementary Fig. 4c–e). Except for *SOX2, CXCL14, KCNH6* and *BRIP1*, none of the enriched human type II hair cell genes (*CALB1, CSRP2, HTRA1, MORC1, SCNN1A, GNAS, NDUFA4L2, B3GNT10, CHRNA1*, Fig. 3e) were previously reported to be enriched in mouse type II hair cells[47,48]. Similarly, the established mouse type II hair cell markers *ANXA4* and *MAPT*[45,46] were not enriched in the human type II hair cell cluster (Fig. 3c). Unexpectedly, the mouse type I hair cell marker, *CALB1*[50] (Supplementary Fig. 4f–h), was enriched in human type II hair cells (Supplementary Fig. 4b). We next quantified the degree of correlation among transcriptomes of human and mouse utricles and found significant, albeit modest, correlation among type I hair cells (0.729), type II hair cells (0.628), and supporting cells (0.829) (Supplementary Fig. 4e). Together, these data suggest that the transcriptome profiles of utricular hair cells and supporting cells only partially overlap between mice and human.

## Diversity of supporting cell subtypes

Differential gene expression results show several pan-supporting cell markers (Fig. 2c, e), of which we have validated *GFAP* and *ANXA2* (Fig. 2f). The mouse and human utricles are organized spatially into the central striolar and peripheral extrastriolar zones (Fig. 4a)[24] with distinct gene expressions described in mice[47,51]. The striolar region may be more critical for utricular function as mutant mice missing this domain exhibit abnormal vestibular responses[51]. Differential gene expression shows 26 genes enriched in striolar supporting cells and 31 genes in extrastriolar supporting cells (Fig. 4b, c). *TECTB*, which marks striolar supporting cells in mouse and chicken utricles[46], was also enriched in human striolar supporting cells (Fig. 4c). We validated *SFRP2* and *FRZB* in sections of utricular sensory epithelia and found that striolar supporting cells robustly expressed *SFRP2* whereas extrastriolar supporting cells expressed *FRZB* (Fig. 4f–f'''), as predicted by computational analyses (Fig. 4c–e, g). Interestingly, although *GFAP* was not found differentially expressed computationally, anti-GFAP staining was relatively more robust among extrastriolar supporting cells (Fig. 4f''–f''', g). These results confirmed the presence of two transcriptionally and spatially distinct human utricular supporting cell subtypes.

## Hair cell degeneration in vestibular schwannoma utricle

Postmortem studies on utricles from vestibular schwannoma patients have reported hair cell degeneration[17,23]. To independently verify these results and determine whether supporting cells mount a regenerative response, we first immunolabeled utricles for hair cells (MYO7A⁺) and supporting cells (MYO7A⁻ and SOX2⁺) from organ donor (*n* = 5, 4 males and 1 female, aged 42.0 ± 25.5) and vestibular schwannoma subjects

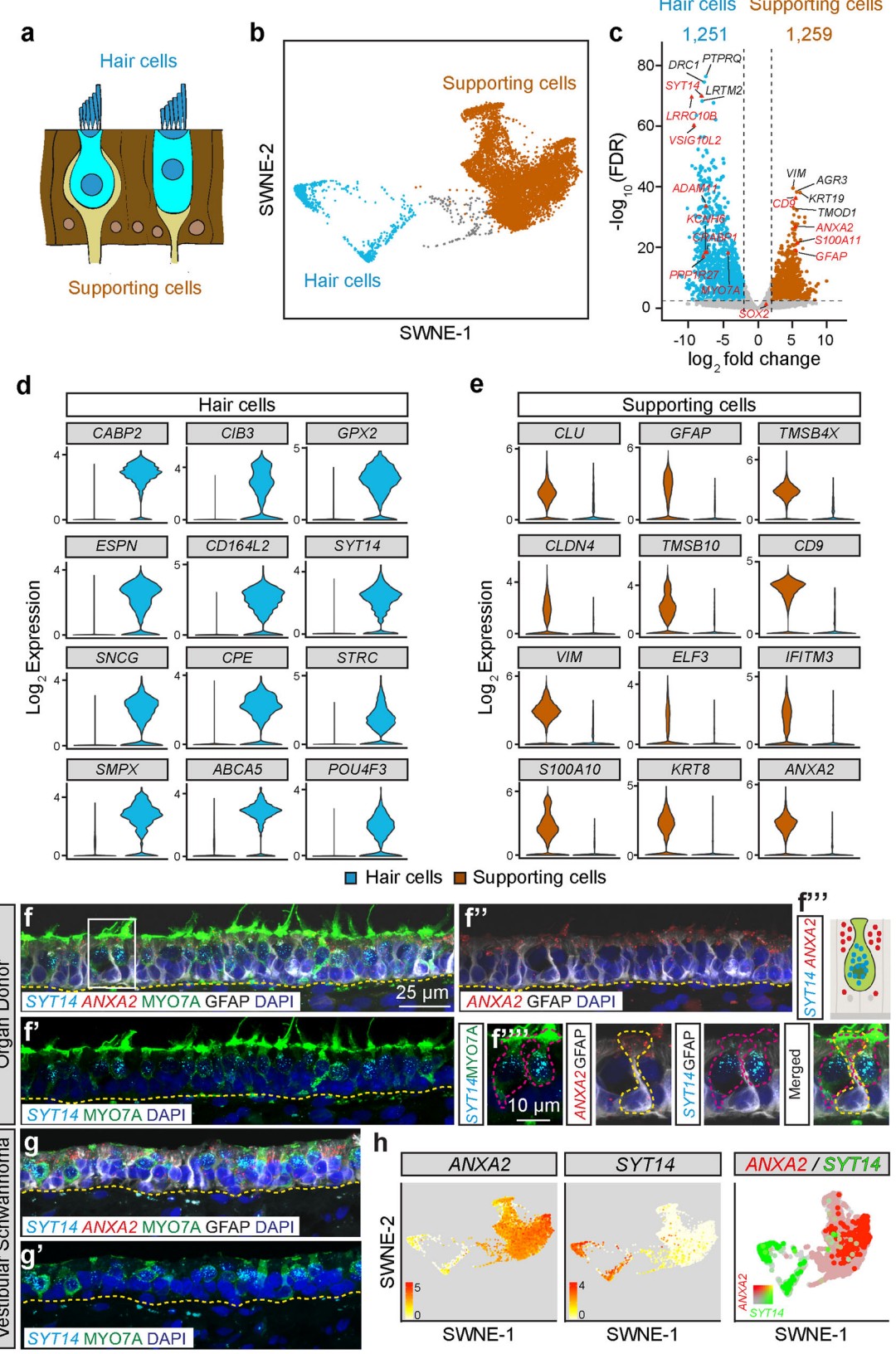

(*n* = 13, 9 males and 4 females, aged 50.5 ± 16.7). As a secondary control, we included utricles from cadavers (*n* = 4, 2 males and 2 females, aged 75.8 ± 5.9, Supplementary Data 1) without any history of auditory and vestibular deficits. Compared to organ donors or cadavers, vestibular schwannoma utricles contained significantly fewer hair cells (-78% fewer, *p* < 0.001, Fig. 5a–d, Supplementary Fig. 5a–b", d–g) and

supporting cells (-21% fewer, *p* < 0.05, Supplementary Fig. 5a'–c). Hair cell counts from individual vestibular schwannoma patients were comparable despite variability of age, gender, laterality, tumor size, or history of radiation in the cohort (Supplementary Fig. 5f, Supplementary Data 1). We found no significant correlation among age, tumor size, auditory function with counts of hair cells or supporting cells

**Fig. 2 | Differential gene expression in human vestibular hair cells and supporting cells. a** Schematic of utricle sensory epithelium highlighting hair cells (blue) and supporting cells (brown). **b** SWNE plot colored by hair cell and supporting cell groups with the hair cell precursor group (gray) excluded. **c** Volcano plot with enriched hair cell versus supporting cell genes using pseudobulk *DESeq2* analysis. The top 5 differentially expressed genes are labeled as well as validated markers (red triangles). Using a log2 threshold of 2 and FDR < 0.01, 1259 genes were enriched in the supporting cells and 1251 in the hair cells (listed in the Source Data file). **d**, **e** Violin plots depicting 12 highly enriched hair cell (**d**) and supporting cell genes (**e**) using the Wilcoxon rank sum test. **f–g'** Validation of the hair cell marker *SYT14* (cyan) and supporting cell marker *ANXA2* (red) in utricles from organ donor and vestibular schwannoma patients. Cryosections were processed for fluorescent in situ hybridization and immunostaining for MYO7A (green), GFAP (gray), and DAPI (blue). Yellow dashed lines mark the basement membrane of the sensory epithelium. *SYT14* is expressed in MYO7A⁺ hair cells in (**f'**) and *ANXA2* is expressed in the cytoplasm of GFAP⁺ supporting cells in (**f''**) in OD. **f'''** Cartoon depicting the expression pattern of *SYT14* (perinuclear cytoplasm) and *ANXA2* (apical cytoplasm) in hair cells and supporting cells, respectively. Boxed area is magnified in individual panels in (**f''''**). Maroon dashed lines outline the MYO7A⁺/GFAP⁻ hair cells expressing *SYT14*. Yellow dashed lines outline the MYO7A⁻/GFAP⁺ supporting cells expressing *ANXA2*. **g, g'** *SYT14* marks hair cells and *ANXA2* supporting cells in vestibular schwannoma utricle, which is notably disorganized with loss of hair cells. **h** SWNE plots displaying enrichment of *ANXA2* in supporting cells and *SYT14* in hair cells. Log₂ expression is shown with 0 in white and maximum in red at the indicated thresholds. Merged color SWNE plot demonstrates minimal overlap between these marker genes. Scale bar = 25 μm in (**f–f''**, **g, g'**) 10 μm in (**f''''**). Source data are provided as a Source Data file.

among vestibular schwannoma patients (Supplementary Data 1). Collectively, these results further indicate that vestibular schwannoma utricles represent diseased organs whereas organ donor and cadaveric organs serve as normal controls.

To characterize the degree of degeneration, we immunolabeled sections of organ donor and vestibular schwannoma utricles for the hair cell marker MYO7A and supporting cell marker GFAP. Compared to organ donor utricles, vestibular schwannoma utricles contained fewer hair cells and the epithelium appeared thinner, resulting in less lamination of hair cell and supporting cell nuclei (Fig. 5e, f). Hair cells in the vestibular schwannoma sensory epithelium appeared short and round, and many lacked hair bundles (Fig. 5e, f). Using TUJ1⁺ calyces or SOX2 protein expression as markers of type I and II hair cells, respectively, we found that vestibular schwannoma organs had significantly fewer of both type I and II hair cell subtypes than organ donor tissues. However, there was disproportionally more type I hair cells in the vestibular schwannoma than organ donor utricles (3.51 versus 0.99, respectively, Fig. 5g, h, Supplementary Fig. 5h, i), suggesting a previously unrecognized, preferential loss of type II hair cells.

Similar to previous observations[38,52], we observed cytocauds, which are actin-rich cables in degenerating hair cells in vestibular schwannoma but not organ donor utricles, suggesting ongoing hair cell degeneration in the former (Fig. 5i, j, Supplementary Fig. 5j, j'). While phalloidin-labeled bundles were apparent in most MYO7A⁺ hair cells in the organ donor utricle, few bundle-bearing hair cells were observed in the vestibular schwannoma utricles (10.7% of 1473 hair cells from 3 vestibular schwannoma utricles, Fig. 5k–m). Remnant hair bundles appeared damaged (splayed) or short (13.4% and 56.1% of 157 hair cells from 3 vestibular schwannoma utricles, Fig. 5m). This contrasts with the organ donor utricles where 89.3% of hair cells displayed bundles, with most appearing long (61.1%, n = 236 hair cells from 1 utricle, Fig. 5k–m, Supplementary Fig. 5k, l). Under scanning electron microscopy, hair bundles with a staircase pattern and packed stereocilia were rarely found (Fig. 5n) in vestibular schwannoma utricles, whereas others lacked stereocilia and kinocilium and instead displayed remnants of short bundles at the apical surfaces (Fig. 5o, p, Supplementary Fig. 5m, o). In other cases, stereocilia within a hair bundle had fused (Fig. 5q). Lastly, we found individual hair cells with two distinct hair bundles located at the opposite poles of the apical surface (Fig. 5q–s), one with taller and thicker stereocilia than the other, possibly corresponding to different maturation stages, perhaps an indication of ongoing repair or regeneration that has been previously suggested and analyzed[38,52]. Together, these results indicate ongoing hair cell degeneration and hair bundle damage in the vestibular schwannoma utricles.

Using our integrated single-cell RNAseq dataset focusing on the sensory epithelium, we performed comparative analysis based on cell types between organ donor and vestibular schwannoma utricles. We focused the comparative analysis to five categories: type I and type II hair cells, hair cell precursors, and extrastriolar and striolar supporting cells (manually grouped among subtypes). For type I hair cells, we identified 1442 genes in both organ donor and vestibular schwannoma (e.g., *MTRNR2L8*, *HSPA1A*, and *HSPA1B* for organ donor and *LGMN*, *RPS4Y1*, and *CAMK2N1* for vestibular schwannoma utricles (Supplementary Fig. 6a)). For type II hair cells, there were a total of 1125 genes in both organ donor and vestibular schwannoma (e.g., *HSPA1A*, *HSPA1B*, and *DNAJB1* for organ donor and *PTGDS*, *SNHG14*, and *EEF1G* for vestibular schwannoma utricles (Supplementary Fig. 6b)). Comparison of hair cell precursor cells shows 1729 organ donor and vestibular schwannoma genes, whereas extrastriolar and striolar supporting cells as a group showed 2183 and 2204 genes, respectively (Supplementary Fig. 6c–e). Overall, this comparison identified genes enriched according to normal (organ donor) or diseased (vestibular schwannoma) conditions (Supplementary Fig. 6). Since there remains remarkable overlap among the transcriptomes of the different sensory epithelial cell types, the significance of these enriched genes is yet to be determined.

## Transcriptomic and trajectory analysis of the human utricle

Both the mouse and human utricle demonstrate a limited degree of hair cell regeneration by supporting cells[7,8,10,13,14], although spontaneous regeneration in vestibular schwannoma utricle in vivo has not been directly observed. To define the molecular underpinning of human hair cell regeneration, we used the *CellRank* algorithm to detect cell-cell transitions during differentiation[53]. We utilized the pseudotime kernel feature of *CellRank* that computes directed transition cell state probabilities. Pseudotime values were obtained using *Palantir*, an algorithm that predicts cell trajectories as a probabilistic process[54]. Using our subsetted sensory epithelial cells, we set the starting point as supporting cells within cluster 1 for pseudotime analysis and used *CellRank* to determine directed cell trajectories (Fig. 6a). Cell-cell transition matrix created by *CellRank* is then projected onto the reduced dimensions SWNE plot to show trajectories of individual cells as streams. Our data shows that the hair cell precursor group is projected to differentiate from a supporting cell state towards the type I and type II hair cell states (Fig. 6a, b), further implicating that supporting cell-hair cell transition occurs in vivo (Fig. 6c). To delineate genes directing supporting cell to hair cell transition and validate the findings from *CellRank*, we created unbiased trajectories using *Slingshot* (Supplementary Fig. 7a, b)[55], focusing on two lineages that originate from supporting cells to type I and type II hair cells via the intermediate hair cell precursors (Fig. 6b, Supplementary Fig. 7b).

Using a generalized additive model fit onto the two lineages of type I and type II hair cells, we identified 9731 and 4315 significant, dynamically expressed genes along the two respective trajectories (Fig. 6d, e). An association test detected five patterns of dynamic gene expression in type I and II hair cell lineages: rapidly upregulated, rapidly downregulated, gradually upregulated, gradually downregulated, and transiently upregulated (Fig. 6d, e). Both the type I and II lineages display rapidly upregulated and downregulated genes (type

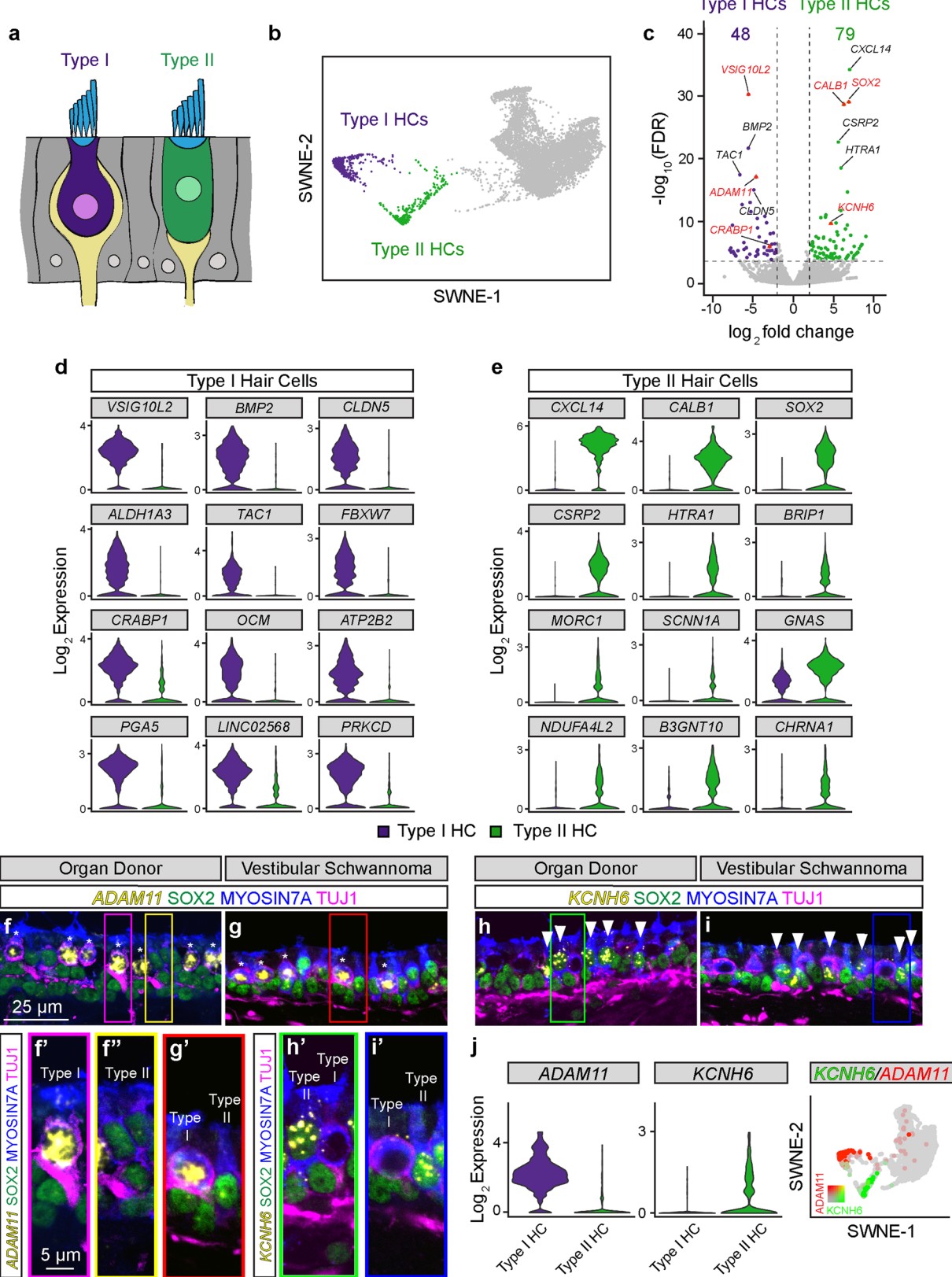

**d** Type I Hair Cells

*VSIG10L2*, *BMP2*, *CLDN5*, *ALDH1A3*, *TAC1*, *FBXW7*, *CRABP1*, *OCM*, *ATP2B2*, *PGA5*, *LINC02568*, *PRKCD*

**e** Type II Hair Cells

*CXCL14*, *CALB1*, *SOX2*, *CSRP2*, *HTRA1*, *BRIP1*, *MORC1*, *SCNN1A*, *GNAS*, *NDUFA4L2*, *B3GNT10*, *CHRNA1*

■ Type I HC  ■ Type II HC

I: *GPX2*, *CIB3*, *GABP2* and *CD9*, *GSN*, *SPARCL1*; type II: *CETN2*, *OTOF*, *ABCA5*, and *AGR2*, *GFAP*, *TMSB4X*, Fig. 6f, f'), implicating shared features of hair cell specification and differentiation. To identify divergent genes among the type I and type II lineages, we computed the Different End Test and identified genes upregulated solely in type I hair cells (*ADAM11*, *VSIG10L2*, and *ESPN*) and type II hair cells (*SOX2*, *CXCL14*, and

*CSRP2*, Fig. 6f"). These results suggest that regeneration of type I and II hair cells share most but not all molecular features.

To characterize mechanisms driving supporting cell-to-hair cell transition, we analyzed dynamically expressed genes in type I and II lineages and found 199 and 185 associated canonical pathways, respectively (Fisher's Exact Test, $p < 0.05$), most of which were shared

**Fig. 3 | Hair cell subtypes in vestibular schwannoma and organ donor utricles.**
**a** Schematic showing amphora-shaped type I hair cells (purple) with apical neck and calyceal innervation. Type II hair cells (green) are goblet-shaped and display bouton-type innervation. **b** SWNE plot with distinct clusters of type I and II hair cells highlighted. **c** Volcano plots of enriched genes in type I and II hair cells. Top 5 differentially expressed genes are labeled with validated genes marked (red triangles). Also see the Source Data file. **d**, **e** Violin plots of enriched markers of type I and II hair cells. **f–i** Fluorescent in situ hybridization and immunostaining validating expression of *ADAM11* (yellow) in type I hair cells (asterisks) and *KCNH6* (yellow) in type II hair cells (arrowheads) in sections of organ donor and vestibular schwannoma utricles counterstained with MYO7A (blue), SOX2 (green), and TUJ1 (magenta). **f'** High magnification image of a type I hair cell, which is SOX2⁻ (green), amphora-shaped, displaying TUJ1⁺ calyx (magenta), and expressing *ADAM11* (yellow). **f''** High magnification image of a type II hair cell from F (yellow box). This

SOX2⁺ (green) type II hair cell is goblet-shaped, displays basolateral process, and lacks ADAM11 expression. **g'** Examples of SOX2⁻ (green), *ADAM11* (yellow) type I hair cells with TUJ1⁺ calyx (magenta), and SOX2⁺ (green) type II hair cells without *ADAM11* (yellow) expression in vestibular schwannoma utricle, respectively, from **g** (red box). **h'** High magnification image of SOX2⁺ (green) type II hair cells with *KCNH6* (yellow) expression, and SOX2⁻ (green) type I hair cells without *KCNH6* (yellow) expression and with TUJ1⁺ calyx (magenta) in organ donor utricle, respectively, from **h** (green box). **i'** Additional examples of SOX2⁺ (green) type II hair cells with *KCNH6* (yellow) expression, and SOX2-negative, TUJ1⁺ calyx (magenta) type I hair cells without *KCNH6* (yellow) expression in organ donor utricle, respectively, from **h** (green box). **j** Violin and merged SWNE plots showing enrichment of *ADAM11* and *KCNH6* in type I and II hair cells, respectively. Scale bar = 25 μm in (**f–i**), 5 μm in (**f'**, **f''**, **g'**, **h'**, **i'**). Source data are provided as a Source Data file.

---

by both lineages. Shared pathways included acute phase response, IGF-1, Interferon, ERK/MAPK signaling, and Wnt/ß-catenin signaling (Fig. 6g). In mice, Wnt signaling have been shown to regulate regeneration of the utricle[11], but the effects on the other signaling pathways on human hair cell regeneration are not known. Moreover, 157 and 139 biological function GO terms were identified for dynamically expressed genes in the type I and II lineages (FDR < 0.05), including ones pertinent to inner ear function (Fig. 6h). These results serve as a mechanistic framework for characterizing naturally occurring human hair cell regeneration.

### Hair cell precursors in vestibular schwannoma utricles

To characterize hair cell precursors, we examined dynamically expressed genes as supporting cells transition into hair cells along pseudotime lineages. As markers of hair cells (Fig. 2), *SYT14* and *MYO7A* were upregulated while the supporting cell marker *ANXA2* was downregulated (Supplementary Fig. 8a). In vestibular schwannoma utricles, where almost all hair cells expressed *SYT14* and *MYO7A* and supporting cells expressed *ANXA2*, we found occasional elongated cells coexpressing *ANXA2*, *SYT14*, and MYO7A, the latter at a relatively lower intensity than in hair cells proper (Supplementary Fig. 8b–b''''), suggesting a transitory cell state between a supporting cell and hair cell in vivo.

We further sought and characterized these elongated, MYO7A-low hair cell precursors by immunostaining for MYO7A and performing in situ hybridization for the hair cell marker *GPX2* (Fig. 7a, b). *GPX2* expression gradually increases along pseudotime similar to *MYO7A* and *SYT14* (Fig. 6f). In sections, *GPX2*⁺, MYO7A-low cells span the basement membrane and the apical lumen of the sensory epithelium (Fig. 7a), unlike MYO7A-high hair cells which are relatively shorter and whose basolateral surface is separate from the basement membrane.

To confirm the presence of hair cell precursors in vivo and to further identify drivers of hair cell regeneration, we examined transcription factors among the identified dynamically expressed genes[56]. A total of 642 significant dynamically expressed transcription factors were identified in both type I and II lineages (Fig. 7b, c). These include *ATOH1*, *POU4F3*, and *GFI1* which are required for hair cell development, survival, and maturation in mice[57–59] (Fig. 7b, c). Next, we found that *GPX2*⁺, MYO7A-low hair cell precursors expressed *ATOH1*, which is typically low or undetectable in hair cells and supporting cells (Fig. 7d, Supplementary Fig. 8c). To differentiate the *GPX2*⁺, MYO7A-low, *ATOH1*⁺ hair cell precursors from *GPX2*⁺, MYO7A-high, *ATOH1* negative hair cells, we measured and found the nuclei distance from the basement membrane of the sensory epithelium to be significantly smaller in the former (Fig. 7e). The nuclei of hair cell precursors are located at a level not significantly different from those of supporting cells, further implicating them as supporting cells transitioning to become hair cells.

Given that *POU4F3* and *GFI1* were rapidly upregulated along the pseudotime trajectories (Fig. 7f), we postulated that they would label both hair cell precursors and differentiated hair cells. As expected, in

organ donor and vestibular schwannoma utricles, all MYO7A⁺ hair cells were marked with anti-POU4F3 and anti-GFI1 antibodies. Strikingly, a subset of SOX2⁺, MYO7A⁻ supporting cells expressed POU4F3 or GFI1 protein and mRNA in both the organ donor and vestibular schwannoma utricles (Fig. 7g, h, Supplementary Fig. 8d–h, i–l), with the latter displaying significantly more POU4F3- (13.98 vs. 1.01%, p < 0.01, Fig. 7i) or GFI1-positive hair cell precursors (12.79 vs. 0.19%, p < 0.05, Fig. 7j). Collectively, these results unveil transcriptomes of human hair cell precursors, which were more prevalent in vestibular schwannoma than organ donor utricles in vivo.

## Discussion

Inner ear sensory hair cells are specialized mechanoreceptors required for hearing and balance functions. Unlike the mammalian cochlea, the vestibular organs have a limited capacity to regenerate lost hair cells[60]. Rodents have served as the primary model system examining mammalian vestibular hair cell regeneration, relying on toxins or transgenic approaches to induce damage[8,10]. Prior studies using tissues procured from vestibular schwannoma patients showed that supporting cells can spontaneously regenerate hair cells after gentamicin treatment in vitro; likewise, ectopic hair cells can form in response to *ATOH1* overexpression[13,14,38]. Here, we procured utricles from organ donor and vestibular schwannoma subjects. The former cohort lacked any history of auditory or vestibular dysfunction, while the latter exhibited hearing loss and/or vestibular dysfunction. Single-cell transcriptomic analysis and in vivo validation experiments revealed conserved transcriptomes of hair cells and supporting cells, despite significant and ongoing degeneration in the vestibular schwannoma organs. Moreover, we found that only a minority of mouse and human hair cell and supporting cell genes overlap, despite significant correlation among those shared genes. By analyzing dynamically expressed genes in the supporting cell-hair cell trajectories, we uncovered markers of hair cell precursors and unexpectedly discovered that significantly more hair cell precursors reside in the vestibular schwannoma than organ donor utricles. Our study reveals candidate mechanisms driving spontaneous human hair cell regeneration and serves as a data-rich resource for research on human auditory and vestibular disorders.

Single-cell transcriptomic analysis has been instrumental in defining cellular heterogeneity in the mouse inner ear as well as discovering candidate mechanisms of development, hair cell regeneration, and pathogenesis of auditory dysfunction[61]. While these studies can serve as foundational models for human disease, investigations on the human inner ear have almost exclusively relied on postmortem, cadaveric tissues[17,18,20–25] or tissues originating from fetuses or diseased inner ears. In the current study, transcriptomes of distinct cell types from the healthy and diseased mature, human inner ear serve as benchmarks guiding future works.

Because the mouse and human inner ear organs share histologic features and constituents essential for hearing and balance functions[62–64], mouse models are commonly used to study human

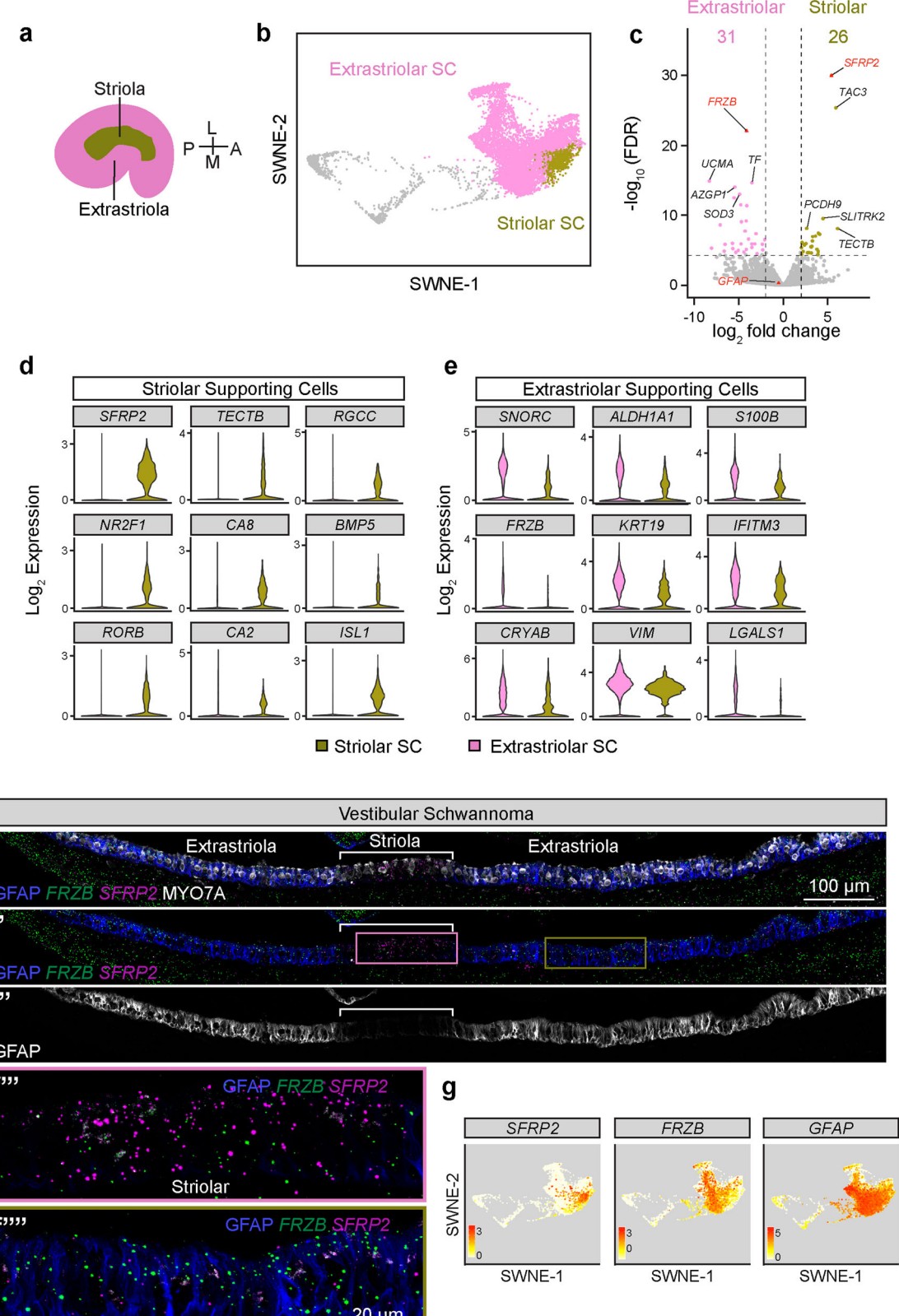

**Vestibular Schwannoma**

auditory and vestibular diseases. Several pathways posited to regulate human vestibular hair cell regeneration (Wnt/ß-catenin, Neurotrophin) were previously linked to regeneration in the mouse vestibular system[11,65], while others have unknown functions (Interferon, IGF-1) (Fig. 6g). When comparing enriched genes between human and mouse utricles[45–47], we found that most genes of hair cells and supporting cells

do not overlap. Such an interspecies difference has previously been suggested in the refined cell subtypes of the mouse and human immune systems[66,67]. Along a similar vein, divergence of neurotransmitter receptors and ion channels is known between mouse and human brains[68]. Such a divergence has implications for translating mouse studies to humans and emphasizes the importance of

**Fig. 4 | Human vestibular supporting cell subtypes in vestibular schwannoma and organ donor utricles. a** Diagram showing the crescent shaped striolar central region and the peripheral extrastriolar region of the utricle (A: Anterior, P: posterior, L: lateral, M: medial). **b** SWNE plot colored by putative striolar and extrastriolar supporting cells. **c** Differential gene expression using *DESeq2* pseudobulk analysis shows 26 and 31 genes significantly enriched in striolar and extrastriolar supporting cells, respectively. See the Source Data file for complete list of genes. **d**, **e** Violin plots showing the expression of select marker genes of striolar versus extrastriolar supporting cell genes using Wilcoxon rank sum test. **f–f″** Combined fluorescent in situ hybridization and immunostaining validating expression of *FRZB* (green) in extrastriolar supporting cells and *SFRP2* (magenta) in striolar supporting cells in sections of vestibular schwannoma utricles counterstained with MYO7A (gray), and GFAP (blue). GFAP (gray) expressed at a high level in extrastriola, but lower in striola (bracket) in (**f″**). **f″**, **f‴** shows high magnification images of striola (pink box) and extrastriola (brown box) in (**f**). *SFRP2* (magenta) is highly expressed in striola, *FRZB* (green), and GFAP (blue) are highly expressed in extrastriola. **g** SWNE plots displaying enrichment of *SFRP2* in striolar supporting cells and *FRZB* in extrastriolar supporting cells. Log$_2$ expression is shown with 0 in white and maximum in red at the indicated thresholds. Scale bar = 100 μm in (**f–f‴**). Source data are provided as a Source Data file.

investigating human auditory and vestibular organs. As we only validated a subset of hair cell and supporting cell genes, the interpretation of other genes of interest may warrant further characterization.

Our study serves as an invaluable resource to study human inner ear diseases. As a case in point, we interrogated the expression patterns of 53 human hearing loss- and 15 vestibulopathy-related genes in our integrated dataset, 8 of which were linked to both conditions. By determining the mean relative expression of these genes, we found that 27 hearing loss genes are uniquely expressed in hair cells and that seven non-sensory cell types highly expressed 38 hearing loss genes (Supplementary Fig. 9a). Among the 20 vestibulopathy-related genes, five are uniquely expressed in hair cells and 11 in seven non-sensory cell types (Supplementary Fig. 9b). In contrast to the rich literature on the pathogenesis of hearing loss genes in hair cells, studies on non-sensory cell types and vestibulopathy-related genes are scarce, and our study establishes a foundation for these future studies.

Inner ear tissues collected from vestibular schwannoma patients have served as the primary model system to study human hair cell regeneration[13,14,38]. Although hair cells are known to have degenerated in inner ear tissues from vestibular schwannoma patients[17,23], these studies on regeneration have employed ototoxins to ablate remaining hair cells in vitro. By comparing vestibular schwannoma to organ donor utricles, we discovered that the former represents damaged tissues with active regeneration in situ and therefore is a distinct model system of human hair cell degeneration and regeneration.

In mice, lineage-tracing experiments show that regenerating hair cells are primarily type II and that they incompletely differentiate[8,10,12]. Our results suggest that regenerating human hair cells display a spectrum of differentiation. While these results may indicate that human vestibular hair cell regeneration, as in mice, requires exogenous factors to enhance the degree of regeneration and differentiation, it is also possible that ongoing damage from vestibular schwannoma modulates these processes.

In summary, using vestibular tissues procured from adult human organ donors and vestibular schwannoma patients, we have revealed the transcriptomes of inner ear sensory and non-sensory cells. We show that vestibular schwannoma utricles harbor both degenerating sensory cells and hair cell precursors. The adult human inner ear has to date remained an enigma at the molecular level, and it is of utmost public health importance to decipher mechanisms of damage and repair. Our results present a foundational resource for future studies on human auditory and vestibular function and diseases.

## Methods

### Ethics and experimental animals
Whole organ utricles from patients with vestibular schwannoma, organ donors, and cadavers were collected under protocols approved by the Institutional Review Board of Stanford University (IRB 27500, 48579, 38993, 50076), Donor Network West's internal ethics committee (IRB #STAN-17−200) and its medical advisory board, University of California, Los Angeles (IRB 10-001449) and Yale University (IRB 2000027777). Written informed consent was obtained from each participant. Participants did not receive compensation for their participation.

Wild-type C57BL/6 J (Jackson Laboratory, #000664) female mice were used. Experiments involving animals were conducted under the animal protocol (#18606) approved by the Animal Care and Use Committee of the Stanford University School of Medicine.

### Human vestibular organ procurement
Whole organ utricles were collected intraoperatively from patients with vestibular schwannoma and organ donors. For vestibular schwannoma patients, utricles were procured from patients undergoing translabyrinthine resection of tumors. Briefly, a neurotologist (R.K.J., P.S.M., N.B., J.C.A., L.S.M., K.V.) opened the vestibule by drilling with a diamond burr on low speed, microdissected out the utricle and removed the attached ampullae. The utricle was then placed in phosphate buffered solution (PBS, pH 7.4; Electron Microscopy Services) on ice until further analysis (<10 min).

Organ donors were referred by Donor Network West (San Ramon, CA). Bilateral utricles were harvested from organ donors as previously described[39,40]. Briefly, a post-auricular incision was made followed by a transcanal approach to expose the middle ear. The tympanic membrane, malleus, and incus were removed while keeping the stapes in situ. To expose the vestibular organs, the bony covering of the vestibule was thinned using a diamond burr on low speed, the stapes footplate removed, and the oval window widened. The utricle was harvested from the elliptical recess and placed in PBS on ice for single-cell RNA sequencing analysis or in 4% paraformaldehyde for histologic analysis.

For cadaveric utricles, temporal bones were collected 10−14 h postmortem, following which specimens were microdissected from temporal bones acquired at autopsy as previously described[19].

### Human subject inclusion criteria
Three cohorts were included in the study. In the first cohort, twenty-four patients (24 ears) with vestibular schwannoma were enrolled from Stanford University and Yale University (Palo Alto, CA and New Haven, CT). Adult subjects with a vestibular schwannoma diagnosed by MRI and hearing loss documented by audiograms were included. Those undergoing a translabyrinthine approach for tumor resection between March 2015 and April 2022 were enrolled into the study. The second cohort included 9 ears from 6 (pediatric and adult) organ donors enrolled through Donor Network West (San Ramon, CA) between January 2018 and July 2020. Both cardiac and brain death donors with no known history of otologic disorders were included. The last cohort consisted of 4 ears from 4 postmortem adult cadavers from University of California, Los Angeles (Los Angeles, California) between February 2015 and January 2017.

### Clinical information
The following was collected for all cohorts: age, gender, and laterality (Supplementary Data 1). For the first cohort, we also collected information pertaining to vestibular symptoms (dizziness, vertigo, or imbalance) or physical signs (positive Romberg sign or unsteady gait), tumor size (longest diameter and its orthogonal measurement in the axial view of the cisternal component), pure tone averages (PTAs) (calculated using 0.5, 1, 2, and 4 kHz), word recognition scores, and history of radiation (Supplementary Data 1). To examine the

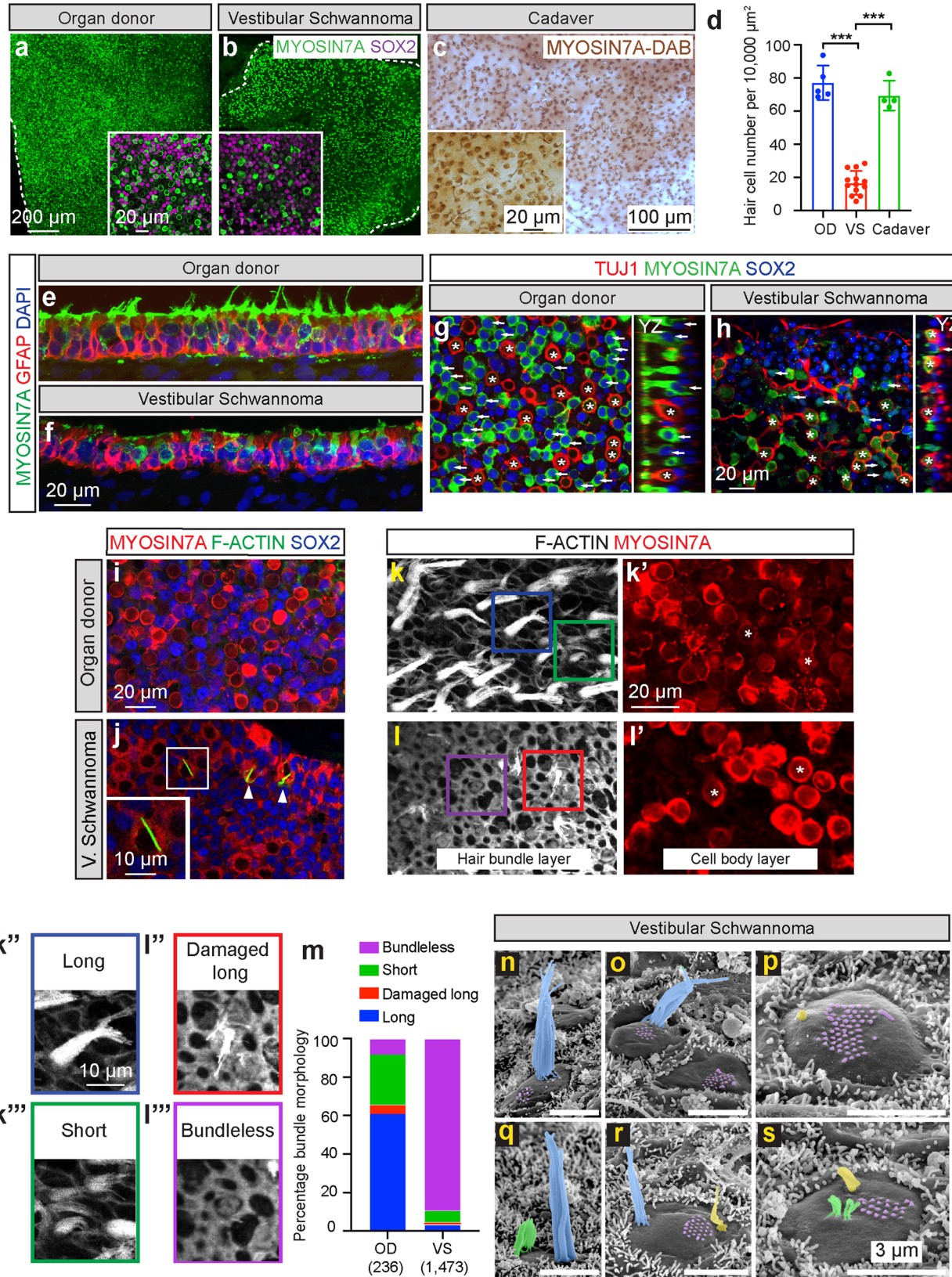

associations between continuous variables and histological results, we used bivariable Spearman's correlation. To compare the histological results between gender and other dichotomous clinical factors (Y/N), we used the Wilcoxon rank sum test. All analyses were performed using the SAS system, version 9.4 (SAS Institute Inc., Cary, NC) (Supplementary Data 1).

**Cryosections**

Tissues were fixed in 4% paraformaldehyde (PFA) (PBS pH 7.4; Electron Microscopy Services) for 16–24 h at 4 °C. After washing in PBS, tissues were cryoprotected in a sucrose gradient (15%, 20%, and 30% for 1 h each), embedded in 100% OCT, and frozen on dry ice. Sections were cut at 10–15 μm and frozen at −80 °C.

**Fig. 5 | Hair cell degeneration in vestibular schwannoma utricles. a, b** Representative low magnification images of MYO7A⁺ hair cells (HC, green) in organ donor (OD) and vestibular schwannoma (VS) utricles, illustrating fewer HCs in the latter. **c** MYO7A-DAB (brown) staining showing many HCs in cadaveric utricles. **d** Quantification showing significantly fewer HCs in VS than OD and cadaveric utricles. **e, f** Compared to OD utricles, the VS sensory epithelium appeared thinner, contained fewer HCs (MYO7A, green), with many supporting cells (GFAP, red) remaining. VS HCs appeared dysmorphic and lacked bundles. **g, h** Relative to OD utricles, VS utricles displayed fewer type I HCs (asterisks, SOX2⁻ (blue), TUJ1⁺ calyces (red)) and type II HCs (arrow, SOX2⁺(blue)). **i, j** F-actin-labeled (green) degenerating HCs (arrowhead and inset, cytocauds with actin-rich cables) in VS but not OD tissues. **k–l′** MYO7A⁺ HCs (red, asterisks) with damaged bundles (F-actin, gray) in VS but not OD tissues. **k″–l‴** Four types of bundle morphology observed in (**k, l**): long (blue), damaged, long (red), short (green), and bundle-less (purple). **m** Percentage of HCs displaying distinct bundle morphology. Most HCs are bundle-less in VS tissues, whereas many OD HCs have long bundles. **n–s** Scanning electron microscopy of VS HCs. **n** HC with intact bundle (blue). **o** Two HCs with remnants of bundles (purple); the left HC has intact bundle. **p** Remnants of stereocilia and kinocilium (yellow). **q** HC with short (green) and tall bundles located at the opposite poles. **r** HC with few intact stereocilia and remnants of a bundle at the opposite pole. **s** HC with a few thin, short stereocilia and remnants of a bundle. Data shown as mean ± S.D. and compared using one-way ANOVA. ***$p < 0.0001$ in (**d**). $n = 13$ for VS, $n = 5$ for OD, $n = 4$ for cadaveric tissues in (**d**), $n = 1473$ cells from 3 VS, 236 cells from 1 OD tissues in (**m**). Scale bar = 200 and 20 μm in (**a, b**), 100 and 20 μm in (**c**), 20 μm in (**e–h**), 20 and 10 μm in (**i, j**), 20 and 10 μm (**k–l‴**), 3 μm (**n–s**). Source data are provided as a Source Data file.

## Immunohistochemistry[12,69]

Utricles from vestibular schwannoma patients and organ donors were fixed in 4% PFA (in PBS, pH 7.4; Electron Microscopy Services) for 40 min at RT or 20 h at 4 °C. Tissues were blocked with 5% goat or donkey serum, 0.1% tritonX-100, 1% bovine serum albumin (BSA), and 0.02% sodium azide (NaN₃) in PBS at pH 7.4 for 1–2 h at room temperature, followed by incubation with primary antibodies diluted in the same blocking solution overnight at 4 °C. The next day, after washing with PBS, tissues were incubated with secondary antibodies diluted in 0.1% tritonX-100, 0.1% BSA, and 0.02% NaN₃ solution in PBS for 2 h at room temperature. After PBS washing, specimens were mounted in antifade Fluorescence Mounting Medium (DAKO) or ProlongGold (Thermo Fisher Scientific) and coverslipped. We used antibodies against the following markers: MYOSIN7A (1:1000-5000; Proteus Bioscience), SOX2 (1:200-500, R&D and 1:400 Santa Cruz Biotechnology), POU4F3 (1:400, Santa Cruz Biotechnology), GFI1 (gift from H. Bellen), GFAP (1:250-1000, Sigma), TUJ1 (1:1000, Neuromics), Osteopontin (1:200; R&D) and CALBINDIN1 (1:500, Cell Signaling Technologies). Secondary antibodies were conjugated with Alexa 488 (1:500, Thermo Fisher Scientific), Alexa 546 (1:500, Thermo Fisher Scientific), Alexa 647 (1:250, Thermo Fisher Scientific), Alexa 594 (1:500, Thermo Fisher Scientific) or Alexa 405 (1:250, Abcam). Fluorescent-conjugated phalloidin (1:1000; Sigma) and DAPI (1:10000; Invitrogen) were used.

Cadaveric temporal bones were fixed in 10% formalin for 6–24 months before microdissection. Utricles were then microdissected, placed individually in a rotary shaker, and incubated for 3 h in a blocking solution containing 2% BSA fraction V (Sigma), 0.1% TritonX-100 (Sigma) diluted in PBS and incubated at 6 °C. Subsequently, the blocking solution was removed, and tissues were incubated with anti-MYOSIN7A in a rotatory shaker for 72 h at 4 °C. Next, tissues were washed with PBS (20 min × 5) and incubated with the secondary biotinylated antibodies (1:1000 in PBS, Vector Labs) for 2 h at RT. Afterwards, tissues were washed again in PBS (20 min × 5) and incubated with the ABC complex (Vector Labs). The antigen-antibody reaction was visualized using 3,3′-Diaminobenzidine (DAB) (Vector Labs). Lastly, utricles were washed with PBS (20 min × 5) and flat-mounted on glass slides and coverslipped with hard mount VECTA-SHIELD solution. Cryosections of mouse utricles subjected to the same protocol were used as positive controls. As negative controls, the primary antibody was omitted or preabsorbed with the antigen and the immunoreaction was performed as described above. No immunoreaction was detected in both types of negative controls.

## Image acquisition and analysis

Images of whole mounts and sections were acquired using confocal microscopy (LSM700 or LSM880, Carl Zeiss Microscopy), Axioplan 2 microscope coupled to a MRC5 (bright field) camera, and using Axiovision AC software (Release 4.8, Carl Zeiss). Z-stack images were taken at 0.5–1 μm intervals. Data was analyzed with Image J64 (Fiji, NIH) and Adobe Photoshop (Creative Cloud, Adobe Systems).

## In situ hybridization

Detection of gene expression was performed using *RNAscope^TM* technology (Newark, CA). Probes are listed in the Source Data file.

Detection of *POU4F3* and *GFI1* in Supplementary Fig. 8 was performed using the V2.5 HD Red *RNAscope^TM* detection system (Cat# 322350, ACDBio). Experiments were performed according to the manufacturer's instructions, with the following modifications: slides were pre-baked at 60 °C for 30 min before tissue processing, boiling was performed for 90 s, and the Protease Plus was diluted 1:3. In some cases, protease digestion was carried out at different temperatures to appropriately balance penetration of probes with retention of immunohistochemical antigens.

Detection of *SYT14*, *ANXA2*, *PPP1R27*, *LRRC10B,* and *KCNH6* in Fig. 2, Supplementary Fig. 2, Supplementary Fig. 4b and 8b was performed using the *RNAscope^TM* Fluorescent Multiplex kit (Cat# 320850). Experiments were carried out according to the manufacturer's instructions (Tissue prep: 320535-TN 11022018, Detection: ACD 320293-UM 03142017), with the following modifications for some tissues to optimize signal quality: boiling time 2.5–5 min, protease time 30 min–1 h, protease temperature 22–40 °C.

Detection of *KCNH6*, *ADAM11*, *FRZB*, *SFRP2*, *GPX2*, *ATOH1*, *CD9*, *CRABP1*, and *VSIG10L2* in Figs. 3, 4, 7, Supplementary Fig. 2, and 4a were performed using the RNAscope Fluorescent Multiplex kit V2 (Cat# 323100) with the Co-Detection Ancillary kit (Cat# 323180). Experiments were carried out according to the manufacturer's instructions (Tissue prep: MK 51-150 RevB 2/11/21 Appendix D, Detection: ACD 323100-USM 2/27/2019), with the following modification: boiling time 2.5–5 min. 20× SCC buffer (AM 9770, Thermo Fisher Scientific) was used to store the slides overnight after probe application.

## Scanning electron microscopy (SEM)

Utricle samples were isolated in washing buffer (0.05 mM HEPES buffer pH 7.2, 10 mM CaCl2, 5 mM MgCl2, and 0.9% NaCl), the otolithic membrane was removed with a brush, and fixed in 4% PFA in 0.05 mM HEPES Buffer pH 7.2, 10 mM CaCl2, 5 mM MgCl2, and 0.9% NaCl for 30 min at RT. The samples were fixed in 2.5% glutaraldehyde and 4% PFA in 0.05 mM HEPES Buffer pH 7.2, 10 mM CaCl2, 5 mM MgCl2, and 0.9% NaCl overnight at 4 °C, then washed, dehydrated in ethanol (30%, 75%, 100%, and 100%, 5 min incubation) and processed to the critical drying point using Autosamdri-815A (Tousimis). Samples were mounted on studs using silver paint and coated with 5 nm of Palladium (sputter coater EMS150TS; Electron Microscopy Sciences). Samples were imaged at 5 kV on a FEI Magellan 400 XHR Field Emission Scanning Electron Microscope at the Stanford Nano Shared Facilities.

## Cell quantification

Using utricles from vestibular schwannoma patients and organ donors, cells were quantified from z-stack images of 25,600 μm², then normalized to 10,000 μm² using Image J64 unless otherwise stated. Images were taken from 1–8 representative areas from the whole sensory

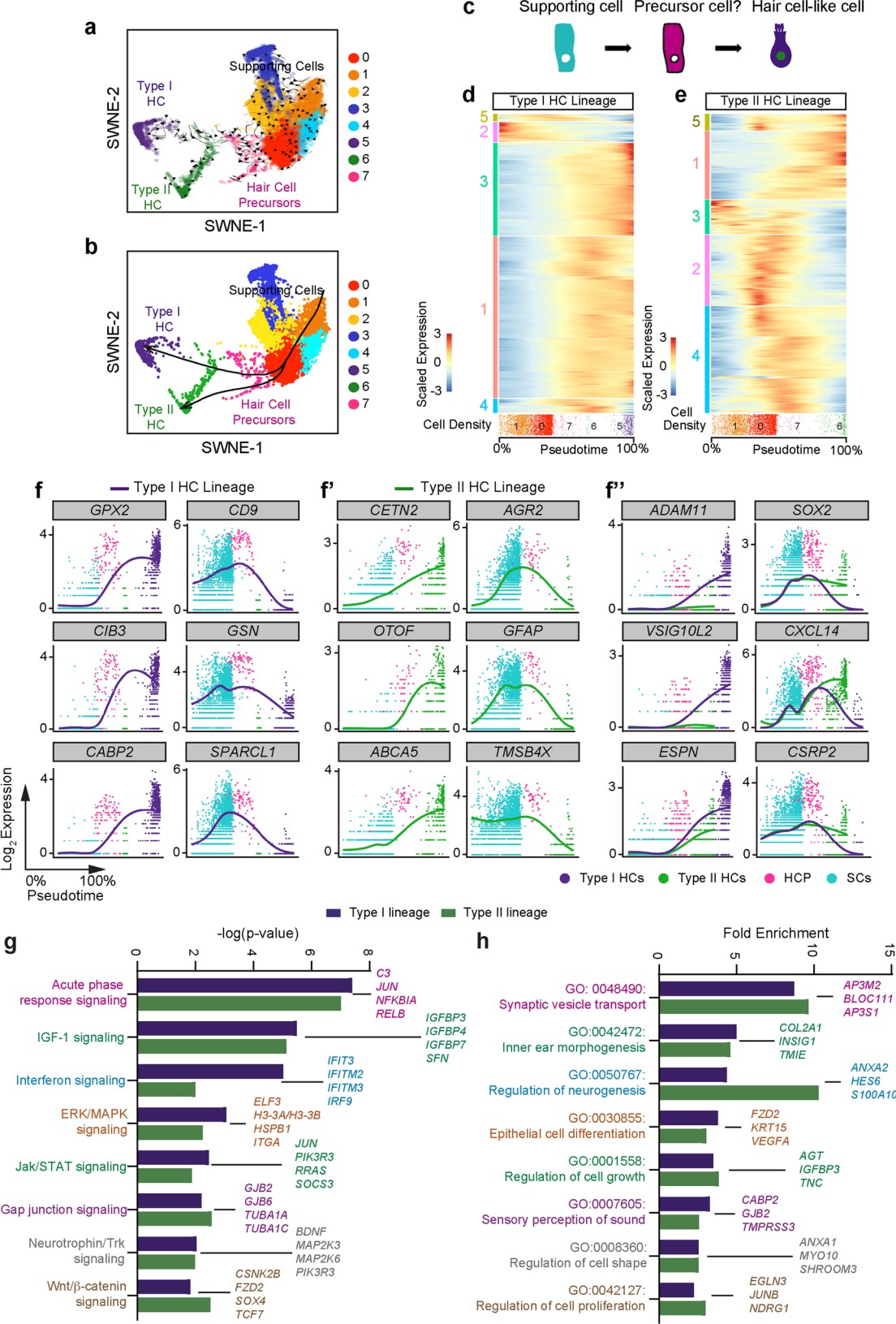

epithelium for analyses. For cadaveric utricles, cells were quantified from images of 10,000 µm². For hair cell precursor cell nuclei quantification, *ATOH1*+ cells that had low expression of cytoplasmic *GPX2*+ were identified. Epithelium was rotated to be parallel to image frame, and the distance from the center of each nucleus to the basement membrane was measured. The nuclei of the immediate surrounding

hair cells (MYO7A+) and supporting cells (GFAP+) were measured in the same way.

**Single-cell isolation**

Utricles from vestibular schwannoma patients or organ donors were placed in DMEM/F12 (Thermo Fisher Scientific/Gibco, 11-039-021) with

**Fig. 6 | Trajectory analysis predicts supporting cell to hair cell transition.**
**a** *CellRank* cell-cell transition matrix trajectory plot showing hair cell precursors differentiating towards mature hair cells (most arrows pointing to the left). **b** SWNE plot with projected type I and type II hair cell lineages and predicted hair cell precursors (cluster 7, pink). **c** Diagram showing supporting cells transition to hair cell-like cells through hair cell precursors. **d**, **e** Heatmap depicting dynamically expressed genes along the type I and II hair cell lineages (9731 and 4315 genes, respectively, FDR ≤ 0.01). There are five patterns of dynamic expression with some genes upregulated, some downregulated, and others transiently expressed in both type I and II lineages. (**d:** 1, 3 = up, 2 = down, 4, 5 = transient; **e:** 1, 5 = up, 2, 4 = transient, 3-down/transient) (colors correspond to the Source Data file). The density of cells, number of cell clusters, and corresponding location along the lineage is depicted below the heatmap. **f**, **f'** Plots showing the dynamic expressions of genes from (**d**, **e**) that increase (left column) or decrease (middle column) along

pseudotime. The individual dots in each plot represent single cells (cyan represents supporting cells, magenta represents hair cell precursors, purple and green showing type I and type II hair cells, respectively) with the x-axis representing pseudotime and the y-axis showing $\log_2$ expressions. The solid line represents the generalized additive model fit to the expression pattern for each gene. **f'** Comparison of type I versus type II hair cell dynamic gene expressions. Using the different end test, we detected statistically significant genes that begin at similar expression levels and end at different levels. *SOX2*, *CXCL14*, and *CSRP2* expression increase in type II hair cells, while *ADAM11*, *VSIG10L2*, and *ESPN* increase in type I hair cells. **g** Graph showing eight top scoring canonical pathways associated with the type I and/or type II lineages along with the corresponding percentage of significantly expressed genes relative to the total number of genes in each pathway. **h** Graph of top biological function GO terms associated with type I and type II lineages. Source data are provided as a Source Data file.

5% FBS during transport to the laboratory. Tissues were then washed twice in DMEM/F12 and any debris and bone microdissected away. The whole utricle was then incubated with thermolysin (0.5 mg/mL; Sigma–Aldrich, T7902) for 45 min at 37 °C prior to mechanical separation of the sensory epithelium from underlying stroma. Next, the whole utricles or the sensory epithelium were digested using Accutase (Thermo Fisher Scientific, 00-4555-56) for 20 min at 37 °C and single-cell suspension was obtained by trituration using a 1 ml pipette. Single-cell suspension was achieved using a 40 µm filter. DMEM/F12 media with 5% FBS was used for this step. Cells were then centrifuged at 300 rcf for 5 min at 4 °C. The supernatant was removed, and cells were resuspended in media. Number of cells per microliter was quantified using a hemocytometer.

The single-cell suspension was then loaded onto a 10× Chromium controller (10x Genomics, Pleasanton, CA) using manufacturer's recommended protocols at the Stanford Functional Genomics Facility. Following cDNA amplification and library prep, the sample was loaded onto an Illumina HiSeq4000 for sequencing. Two vestibular schwannoma samples were separately processed as described above using two separate 10x Genomics captures and subsequently sequenced in separate lane. One human organ donor sample was also processed as above, and the sample was split into two 10x Genomics capture runs and sequenced on two separate lanes.

### Data preprocessing and quality control
Standard 10x Genomics *CellRanger* pipeline was utilized for demultiplexing, alignment, and quantification. Filtered count matrices for each capture (8 total) were then loaded into R to create a *SeuratObject*. We used the *DoubletFinder* method to identify and exclude doublets from eight separate 10x captures with a total of 2240 doublets detected using default parameters[70]. Cells with at least 300 features and with less than 20% mitochondrial genes were kept (Supplementary Fig. 1d). Cells from two vestibular schwannoma utricles were excluded because they displayed >20% mitochondrial genes. The vestibular schwannoma *SeuratObjects* were then combined, and the organ donor *SeuratObjects* were combined.

### Normalization and data integration
Seurat's reciprocal PCA and reference integration method was used to integrate the 8-cell captures post quality control. The NormalizeData function was first employed followed by the top 2000 highly variable genes for each condition (organ donor and vestibular schwannoma). Data was then scaled using the ScaleData function and integration anchors identified using the reciprocal principal component analysis reduction in *Seurat* v3[71]. The integrated data object was then scaled using ScaleData function in *Seurat*. For integrated comparison of organ donor and vestibular schwannoma sample cell subtypes, *Seurat*'s default pipeline was used where the FindMarkers function was used to compare the same cell type between the two conditions. For separate analysis, the organ donor and vestibular schwannoma samples were

merged as above and taken through the entire outlined pipeline separately.

### Cell clustering
Principal component analysis (PCA) was performed using the first 20 PCAs. Initial standard UMAP was used for data visualization, followed by clustering with a resolution of 0.2. The *chooseR* algorithm was used to identify the optimal resolution of 0.2 for cell clustering[42] (Supplementary Fig. 2b–b"). We represent our data using UMAP as well as the previously described Similarity Weighted Non-negative Embedding (SWNE) method for preserving both local and global data structure[41]. The SWNE *Seurat* wrapper function was used per default parameters.

Cell annotation was carried out based on marker genes as described in the literature[72–76]. To focus our analysis on the sensory epithelium, we excluded certain cell clusters (dark cells, stromal cells, melanocytes, vascular cells, roof cells, macrophages, immune cells, and Schwann cells). The remaining cells were then reclustered and used for further downstream analysis (Supplementary Fig. 2). The *chooseR* algorithm was run again following subsetting to arrive at a clustering resolution of 0.2.

### Differential gene expression
The Wilcoxon rank sum test from *Seurat*[71,77] was used for identification of differentially expressed genes among the different cell groups with default parameters that utilized the Bonferroni correction for false-discovery (FDR < 0.01). Cell clusters were defined based on known marker genes for each cluster. Only positive marker genes were kept in the differential gene expression analysis. When comparing two groups of cells (Figs. 2, 3, 4), we used pseudobulk analysis using the *DESeq2* algorithm. Each 10× capture was considered as a separate run, generating 4 replicates that were used for pseudobulk analysis purposes. For example, there are hair cells from two organ donor and four vestibular schwannoma utricles that were compared generating a robust differential gene expression comparison. For generation of volcano plots, *DESeq2* results were plotted using *EnhancedVolcano* with indicated thresholds.

### Trajectory analyses
To identify candidate hair cell precursors, we focused our analysis on the sensory epithelial cells and known hair cell, and supporting cell genes. *Seurat* clustering identified five supporting cell subtypes, two types of hair cells, and one group of putative hair cell precursors. We first reprocessed our data to work more easily in a Python environment. The raw fastq files generated from Illumina sequencing were aligned using *STARsolo* on a local high performance computing cluster (ACCRE – Vanderbilt University)[78]. These were combined to create an anndata object for analysis in Python. Eight unfiltered anndata objects were then loaded into a *Jupyter* notebook. They were filtered against the barcodes (cells) in the finalized *Seurat* object containing the SWNE

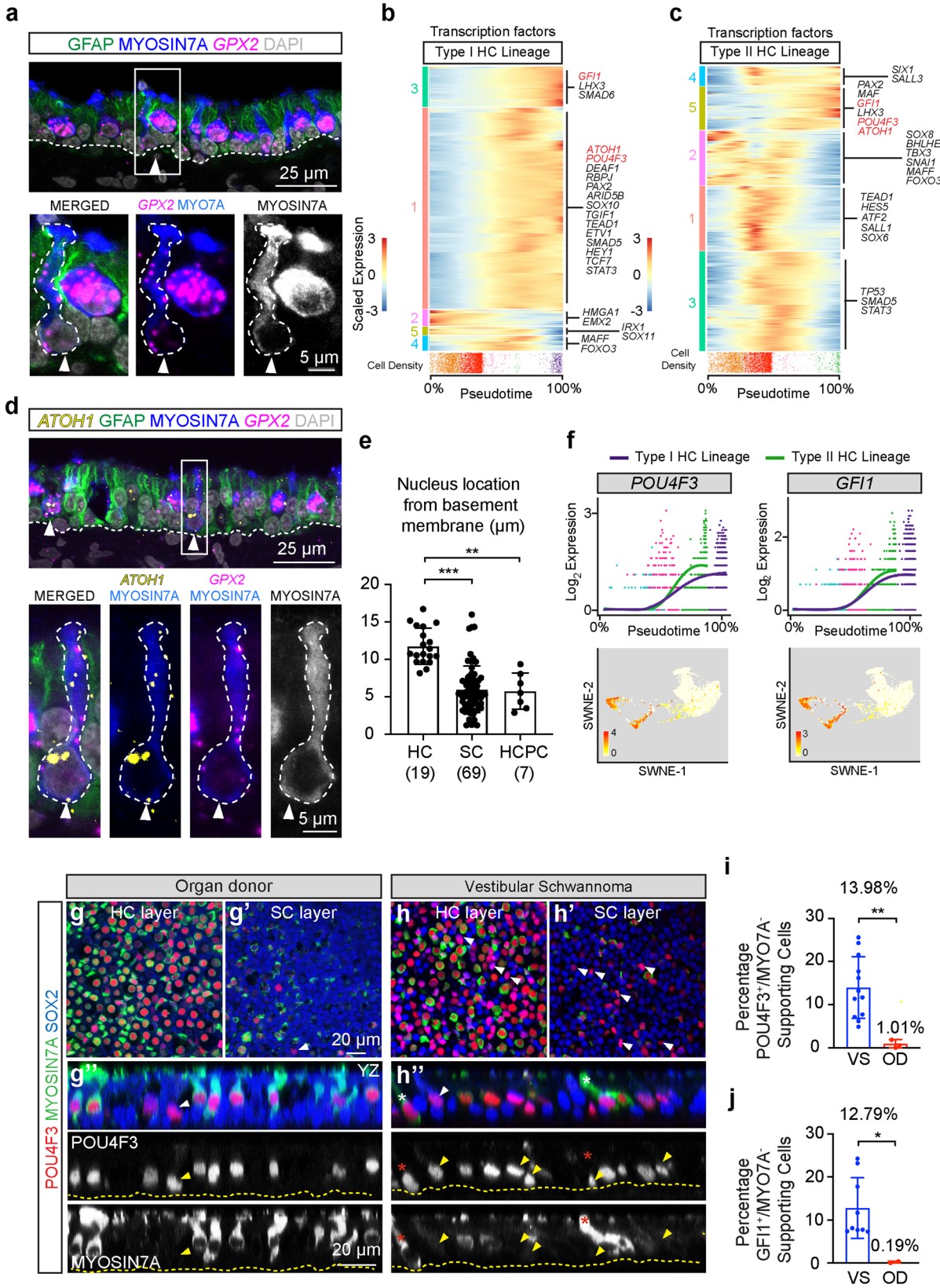

reduced dimensions. The combined anndata object contained the same cell names (barcodes) as the original SWNE *Seurat* object. Next, we used *CellRank's* pseudotime kernel method to calculate the cell-cell transition matrix[53]. First, *Palantir* trajectory analysis was used on the SWNE plot with a start cell in cluster 1[54]. This generated pseudotime data that were used in *CellRank's* Pseudotime Kernel function. The cell-

cell transition matrix directions were then projected onto the SWNE plot (Fig. 6a).

To validate the *CellRank* cell-cell transition matrix direction, we used another trajectory analysis algorithm for further analysis. *Slingshot* was used to connect the cell states on the reduced dimensions as visualized by the SWNE plot (Supplementary

**Fig. 7 | Regeneration in human utricles. a** An elongated hair cell precursor cell (HCPC, arrowhead) expressing a low level of *GPX2* (magenta), MYO7A (blue), and DAPI (gray). Dashed line marks the basement membrane (BM). Magnified image showing a *GPX2*/MYO7A⁺cell, whose nucleus is lower than other HCs. **b, c** Five patterns of dynamic expression of transcription factors along the type I and II hair cell lineages: some upregulated, some downregulated, and others transiently expressed in both type I and II lineages. (**b:** 1, 3 = up, 2, 5 = down, 4 = transient, **c:** 1, 3 = transient, 2-down/transient, 4, 5 = up) (validated markers highlighted in red. Also see the Source Data file). The location of cells along pseudotime for each lineage is depicted as cell density below the heatmaps. **d** shows elongated HCPCs (arrowhead) expressing *GPX2* (magenta), *ATOH1* (yellow), MYO7A (blue), and DAPI (gray) with a nucleus close to the BM. **e** Nuclei of HCPCs are significantly closer to the BM that those of HCs and supporting cells. **f** Dynamic expression and SWNE plots predicting upregulation of both *POU4F3* and *GFI1* as supporting cells transition to both type I and II HCs, with HCPCs colored in magenta. **g**–**h"** VS tissues contained many more POU4F3⁺/SOX2⁺/MYO7A⁻ (red/blue/green) HCPCs (arrowheads) than OD tissues. Orthogonal views show POU4F3⁺/SOX2⁺/MYO7A⁻ cells (arrowhead) with nuclei near the BM. Occasional elongated POU4F3⁺/SOX2⁺/MYO7A⁻ cells (asterisks) were found in VS utricle (**h"**), with nuclei near the BM and below that of other HCs. **i** The percentage of POU4F3⁺/MYO7A⁻ supporting cells was significantly higher in VS than OD tissues. **j** Percentages of GFI1⁺/MYO7A⁻ supporting cells are significantly higher in VS compared to OD tissues. Data shown as mean ± S.D. and compared using two-way Student's *t*-tests or compared using Kruskal–Wallis with Dunn's multiple comparisons. \*\**p* = 0.0016 between HCs and HCPCs, \*\*\**p* < 0.0001 between HCs and SCs in (**e**), \*\**p* = 0.0091 in (**i**), \**p* = 0.0384 in (**j**). *n* = 19 HCs, 69 SCs, and 7 HCPCs from 3 VS tissues in (**e**). *n* = 12 for VS, 3 for OD tissues in (**i**), *n* = 9 for VS, 2 for OD tissues in (**j**). Scale bar = 25 and 5 μm in (**a, d**), 20 μm in (**g**–**h"**). Source data are provided as a Source Data file.

Fig. 7a)[55]. The reduced dimensions as captured by SWNE was used to build this trajectory using the *Slingshot* algorithm[55] (Fig. 6b, Supplementary Fig. 7a). We defined the starting point as cluster 1, and endpoints as the two hair cell types. Unbiased lineage trajectory modeling identified four different lineages (Supplementary Fig. 7b) using the default 6 knots within the *tradeSeq* package[79]. We chose to focus our analysis on the two lineages that generated type I and type II hair cells (Fig. 6b). A generalized additive model (GAM) was then used to fit the dynamic expression of each gene along each lineage using the fitGAM function in *tradeSeq*. Dynamically expressed genes were identified along each lineage using the AssociationTest function. Significantly associated genes were determined using an FDR ≤ 0.01. The diffEndTest function in *tradeSeq* identified genes with dynamic expressions that significantly diverge following a common test expression level (Fig. 6f). The plotSmoothers function allowed the plotting of expression of select individual genes along pseudotime.

### Pathway analysis
Genes that passed *tradeSeq* association test, which assesses significant changes in gene expression as a function of pseudotime, for either type I or type II hair cell lineages with an FDR ≤ 0.01, were uploaded into Ingenuity Pathway Analysis (IPA, QIAGEN Inc., https://digitalinsights.qiagen.com/IPA) for core analysis[80]. IPA was performed to identify canonical pathways that are most significant to type I and type II hair cell lineages. Gene Ontology was performed using DAVID to reveal significant biological processes[81,82].

### Statistics & reproducibility
Statistical analyses were conducted using Microsoft Excel (Microsoft) and GraphPad Prism 7.0 software (GraphPad). Ordinary one-way or two-way ANOVA with Tukey's multiple comparisons, Kruskal–Wallis with Dunn's multiple comparisons, or unpaired student's *t*-tests were used to determine statistical significance. *P* < 0.05 was considered significant. Data shown as mean ± S.D. Unless otherwise indicated, experiments were replicated 3 or more times. The experiments were not randomized, and the investigators were not blinded to experimental groups during data analysis.

### Reporting summary
Further information on research design is available in the Nature Portfolio Reporting Summary linked to this article.

## Data availability
All relevant data and source data are included in this article and Supplementary Information. Source data are provided with this paper. The single-cell RNA sequencing data generated in this study have been deposited in NCBI Gene Expression Omnibus database under accession code GSE207817. The processed data has further been deposited in a user-friendly manner in the gEAR portal, which is a website for visualization and analysis of multiomic data under the following URL: https://umgear.org/p?l=human-utricle-sc-atlas. Source data are provided with this paper.

## Code availability
The following publicly available algorithms and software were used for data analysis presented in this manuscript: DoubletFinder (v2.0.3), scater (v1.18.6), chooseR (helper functions found at: https://github.com/MenonLab/chooseR), Seurat (v4.0.2), tradeSeq (v1.5.07), slingshot (v1.8.0), pheatmap (v1.0.12), swne (v0.6.14), Palantir (v1.3.0), CellRank (v2.0.0), Ingenuity Pathway Analysis, and DAVID. Publicly available vignettes and code were used as well as default settings unless otherwise stated in the text. There was no custom code or algorithms generated.

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

## Acknowledgements

We thank our laboratory for insightful comments on the manuscript, J. Oghalai, W. Dong, J. Burns, D. Ellwanger, J.P. Cartailler, T. Cheng and gEAR team for excellent technical support, Y. Ma for statistical analyses, H. Bellen for sharing reagents, and A. Salehi and staff at Donor Network West and Stanford Otolaryngology surgeons for assistance with tissue procurement. This work was supported by NSF ECCS-2026822 (the Stanford Nano Shared Facilities), S10OD018220, S10OD021763 (Stanford Functional Genomics Facility), the Lucile Packard Foundation for Children's Health, Stanford NIH-NCATS-CTSA UL1 TR001085, Child Health Research Institute of Stanford University, NIDCD/NIH R21DC015879, National Natural Science Foundation of China 81670938 and 82071056 (T.W.), Stanford University Medical Scholars Research Program, Howard Hughes Medical Institute Medical Fellows Program, NIDCD/NIH F30DC015698 (Z.N.S.), American Hearing Research Foundation, R01DC016318 (D.N.), U24DC015910 (A.I. and I.A.L.), R01DC016409, R01AG081608, R21DC019457, the Stanford Maternal and Child Health Research Institute (N.G.), K08DC019683, Hearing Research, Inc., VUMC Discovery Scholars Program (T.A.J.) and R01DC016919, R01DC013910, R01DC020879, R01DC021110, T32DC015209, K24DC020986, U24DC020857, Department of Defense MR130316, Akiko Yamazaki and Jerry Yang Faculty Scholar Fund, and California Initiative in Regenerative Medicine RN3-06529 (A.G.C.), and generous support by the Yu, Ogawa, and Donoho families and the Bill and Susan Oberndorf Foundation.

## Author contributions

T.W., S.E.B., D.K.H., G.S.K., I.A.L., N.G., T.A.J., A.G.C. designed experiments. D.K.H., G.S.K., Z.N.S., K.A.A., Y.V., J.C.A., L.S.M., P.S.M., N.H.B., R.K.J., J.K., A.I., I.A.L., T.A.J. procured tissues. T.W., A.H.L., N.P., D.W., S.E.B., N.G., D.N., T.A.J., I.A.L., M.S., A.G.C. performed experiments. T.W., A.H.L., S.E.B., G.S.K., P.J.A., R.Z., N.G., S.H., T.A.J., A.G.C. analyzed data. T.W., A.H.L., S.E.B., P.J.A., N.G., T.A.J., A.G.C. prepared the manuscript.

## Competing interests

The authors declare no competing interests.
