## [Peer Review File · Nature Communications]

Single-cell transcriptomic atlas reveals increased regeneration in diseased human inner ear balance organsREVIEWER COMMENTS

Reviewer #1 (Remarks to the Author):

This manuscript by Wang et al., describes single cell transcriptome analysis of adult utricle from “healthy” donor and patients with a vestibular schwannoma, using 10X pipeline and Seurat package for analysis. They further make use of vestibular samples from organ donors and patients with vestibular schwannoma to confirm few markers expression and hair cells morphology in situ. They finally use slingshot and RNA velocity for trajectory analysis to suggest the presence of hair cells precursors in their dataset and to draw some molecular trajectories between supporting cells and hair cell types.

One main result is the observation, in their single cell data, of 25 clusters of cells, including 2 types of hair cells (type I and type II, as in mice), and possibly 7 cell types or cell states of supporting cells (not confirmed in vivo). The authors describe molecular similarities with published mouse datasets. They also claim divergence; however, previous murine data were during development; moreover, any difference could easily result from the low number of organ donor for the healthy condition (only one), and from the difference in the quality (depth) of the datasets, whether this is in their human data, or in the published mouse datasets. It is therefore too preliminary to conclude any obvious difference. Moreover, the authors should have performed a full comparative between datasets from healthy and schwannoma samples. See specific comments below as well on data analysis and clustering.

Also, previous studies have already demonstrated in vivo morphological abnormalities and death of vestibular hair cells of the utricle from human with vestibular schwannoma. Some have also shown that supporting cells can spontaneously regenerate hair cells in human utricle with schwannoma, and that overexpressing specific transcription factors could induce ectopic hair cell formation. The principal claim of the authors of the present study is to demonstrate regeneration in vivo, and not ex vivo, as previously done. While I acknowledge the interest in studying the molecular regulation of hair cell regeneration, this study does not prove the existence in vivo of hair cell precursor cells (their description here are subjective, not scientifically based), and their potential to regenerate hair cells, which would have been the main impact of this study, should the authors had demonstrated it. Moreover, they do not compare datasets to prove any increased regeneration, and do not properly demonstrate it in situ.

There are multiple short cuts, limited impact in its current form, and many overstatements that preclude publication in this high-profile journal. I strongly suggest improvement of data analysis (+ adding samples), less focus on repeating previously published data (or condense them), and instead confirm the existence of HC precursors in situ (using various, new markers) and their potential (ex vivo) of differentiating into hair cells.

Specific Comments:

Title:

should change “inner ears” to “utricle” or “otolith organs”, as it is not the whole inner ear that has been analyzed.

Intro:

- line 77-78: yes, previous study has shown regenerative capacity of supporting cells, to form new HCs in utricle of schwannoma patients (Taylor RR et al., 2018, eLIFE).

Results:

- line 139: only one donor is highly limiting for single cell transcriptomics, even more when data must be compared with pathological samples. Moreover, from the two donors with vestibular schwannoma, one had vestibular symptoms (uncharacterized?), the other not. This may have large influence on the various characteristics of the tumor, its localization and size, and therefore on the cellular and molecular characteristic of the dissected tissue. How can the author analyze and differentiate that? Also, the average age of the schwannoma donors is about 20 years younger than that of the organ donor (Ctr). Knowing the age dependent cellular and molecular changes in the inner ear epithelia, how can this be taken into consideration in their analysis. With the high number of samples they process for histology, it is unclear to me why they only analyze 1 healthy donor sample and 2 schwannoma samples for single cell RNAseq.

- line 153-159: need references to gene sets. Why TMC1 only observed in cluster 13, not in cluster 17. Also, in Jan et al., 2021, Cell Rep, S100A6 is described as a non-HCs epithelial marker gene (in utricle). Notably, it seems necessary here to better justify the identity of cluster 20 as hair cell precursors.

- line 161: hair cell and supporting cell clusters are somehow supported with expression of few marker genes. But hair cell precursor cluster is defined arbitrarily. There is no data to sustain this claim, which is important for the rest of the study.

- line 162: EPCAM expression is not explained, and not restricted to the selected clusters mentioned. Importantly, one can see in Fig. 1D and E that cluster 17 is not clean, but certainly contaminated, and contain different types of cells, with clear Sox2 and EPCAM co-localizing, and absence in most cells of MYO7A. This suggests the presence of supporting cells within cluster 17. It is important to note also that the use of DoubletFinder is not always sufficient to discard all doublets, or contamination. Moreover, SOX2, a supporting cell marker, is not expressed in clusters 5 and 7, depicted as supporting cells in Fig. S2.

- line 173: without in situ confirmation, this conclusion cannot stand. Clusters do not equal cell types by default, and identity is not confirmed here, but suggested only based on similarities. See above comments for clusters without apparent identities.

- line 179, there is no data, for now, supporting the identity/existence of putative HC precursors. Moreover, sharing few marker genes with HC clusters is not sufficient to claim such identity.

- line 186: need references to previous reports.

- line 198: there is no comparative analysis of Ctr vs Schwannoma samples to claim this. And how are these cells clustered individually? This should be done in Fig. 1, irrespective of this claim.

- line 209, and earlier. Data in tables are presented as fold change; raw data on expression levels of all genes, and in all clusters, should be given first, "fold change" is an analysis.

- line 211-223: considering a possible contamination or clustering biases, as mentioned above, such type I vs type II dichotomy should be taken very carefully. Also, previous work referred here (Burns et al. 2015 and McInturff et al., 2018) concerns developing inner ear, therefore comparative analysis is difficult due to stage differences, more than considering mice versus human.

- line 223: Calb1 is not a type I hair cell markers, but a marker for calyx afferents innervating type I HCs (Stone et al., 2021, J Neurosci + numerous, previous, literature on the subject). The absence of markers do not explain differences between mouse and human, but could

find other, various explanations, such as data quality and analysis (also, developing murine or chicken utricles must be different to any adult samples). The right panel in Fig. 3H might also explain variations, as clustering does not seem clean, as suggested in previous comments.

- Fig 3F: Discrepancy with text line 235; SOX2 staining is found both in type I (F') and type II (F''), and is cytoplasmic, while it should be nuclear. Similarly, KCNH6 is also found in both morphological types in the panels. Specificity in panel 3G is very unclear too.

- line 244: too preliminary data to conclude that, as previously mentioned, this could be due to data quality and analysis, and age.

- Fig. S3A does not clearly establish CRABP1 as a type I hair cell marker. Not obvious neither for VSIG10L2, unclear which hair cell type it labels in the ISH/immunostaining panel.

- line 254 and related figures: How striolar versus extrastriolar supporting cells were identified unbiasedly is not explained. The striolar supporting cell population represents a very small number of cells that do not seem to separate on their graph (Fig. S3D) from the other supporting cells. Extrastriolar cells in S3.F' do not seem to statistically represent a molecularly different group of cells, with no clear marker genes differentiating them from striolar cells. An explanation for this could be that the striolar cells are also found in the extrastriolar SC group in their dataset. If striolar SC would be unbiasedly identified, one would expect to see them relatively separated from (or at the edge of) the extra SC large cluster in S3D, and not surrounded by it. There are no data to substantiate their conclusion on divergence between mouse and human.

- Fig. 4A-D: marker expression could change (decrease) in schwannoma samples, with no consequence on cell number (density). This is visible in Fig. S4A, B, especially in YZ plan of the B panels. To test this, authors should show at DAPI counter-staining, or similar.

Moreover, and in relation to a previous comment, could the authors separate in their scRNAseq analysis and clustering, the schwannoma from Ctr datasets, and analyze differences in expression levels. Can they also then reproduce the clustering of HCs and SCs using the two separated datasets? Anyhow, similar loss of vestibular hair cells and stereociliary abnormalities have been described in past studies in vestibular schwannoma samples (Hizli et al., 2016; Taylor et al., 2015).

- line 291: data are not showing preferential loss of type II hair cells, both cell types are greatly affected, with indeed more for type II. This difference could also be accounted by the difficulty in counting HCs in the schwannoma samples using HC markers. Indeed, the authors use a HC marker (SOX2) to count type II cells (SOX2 levels seem to decrease in a large number of cells in S4B''), while they are using a marker of calyx nerve endings for counting type I cells.

- line 292: similar to observations previously published (Taylor et al., 2015).

- line 309: not only suggested but analyzed ex-vivo (Taylor et al., 2015).

- line 339: this conclusion of a supporting cell-to-hair cell transition is subjective and should be tuned down. Using slingshot as a first method is incorrect since it is biased, the authors choose the trajectory, which however is the first question, and should not be deduced subjectively. Therefore, the authors should perform first velocity and associated package from the S. Linnarsson's lab, for unbiased trajectory inference; I would suggest using two tools as the first approach, they work synergistically, scVelo and CellRank. Then, this can be followed with Palantir or scFATE (from Kharchenko's lab) for actual analysis of the trajectories. Eventually, they can confirm with Slingshot, but other tools are better suited for unbiased analysis.

Data on Fig. 5B and C are not showing 2 trajectories. The changes in gene expression do not delineate dynamic changes that are directional, which would be expected if there is trajectory of cell types/states. Color coding is not complete, with end points not shown (?). For type II for instance, it should show trajectory with following clusters order: 7-4-10-(2)-9-

20-17, and the color coding.

Data on Fig. 5D are not convincing, as they show selected genes only, yet with a discontinuity in their trajectory of expression levels. Also, to appreciate any trajectory, cells should be labelled in the appropriate color code of the cluster they belong, as in Fig. 5A.

Overall, one cannot conclude on an actual differentiation process in vivo. The authors wish to answer whether regeneration occurs in human in vivo, but such gene expression analysis is artificial in essence and must be proven/assessed experimentally using similar ex vivo methods as in Taylor et al., eLIFE, 2015, onto which they test their molecular pathways.

- line 344-346: concluding on the trajectory analysis is incorrect. There is no proven “effects”.

- line 350: Two things should be done for this conclusion. ONE, that the authors prove, at least ex-vivo, the cellular/molecular trajectory, and TWO, that the data (gene expression) and analysis (see above comments for weaknesses in cluster/computational analysis) are of sufficient quality for identifying molecular trajectories and gene-regulatory networks dynamics in details. Unfortunately, both are not shown.

- line 354-362 and Fig. 6G-H: levels of MYO7A (as mentioned above already) could simply decrease in schwannoma samples. Claiming there are more supposedly HC precursors in schwannoma samples is too preliminary to claim. Also, POU4F3 cells with low levels or absence of MYO7A in schwannoma epithelium in 6H” are clearly positioned in the HC layer, strongly suggesting these are HCs which has lost MYO7A. It is very well known (and confirmed in this study) that vestibular HCs degenerate in patients with vestibular schwannoma, it is therefore not surprising, and instead expected, to see downregulation of some HC markers in schwannoma samples. In support, while not being a particular focus in the field, decrease/loss of MYO7A has been shown in cochlea hair cells in pathologic conditions (Fei yu et al., Neurotox Res, 2016).

- Fig. 6, same comments as for Fig. 5. (see above).

- Fig. 6I: definitely needs comparative analysis of scRNAseq data between ctr and schwannoma samples, to show potential absence of cluster 20 in healthy sample. Then comes the issue of the paucity of cell number in cluster 20; only one “healthy” donor is limiting and this could easily explain potential differences, if there would be any. Additional donor samples are necessary.

- line 363-388: data are sometimes suggestive (as with POU4F3), sometimes against (as with ATOH1) trajectory from supposedly precursors to HC types. Anyhow, one cannot conclude in vivo trajectory from the analysis; it is overstated to write “To confirm the presence of hair cell precursors in vivo”. line 374: no computational method can validate such trajectory, only experiments can favor or validate, which has not been done.

Reviewer #2 (Remarks to the Author):

Wang et al. describes single cell RNA seq analysis of human vestibula and discovered a variety of cell types and clusters. Some clusters have both similar and divergent gene expression when compared to mouse vestibula. Many of these genes identified are novel and have been confirmed with in situ hybridization again in human vestibular samples. The most interesting findings are the cluster of “hair cell precursors” in schwannoma utricles that resemble regenerating hair cells. These interesting cell populations provided some insight on transitory pathways and new genes/factors that can play roles in regeneration of hair cells in humans. Their results are solid and convincing, and the manuscript is well written and easy to follow. The difficulty in collecting human vestibular samples for analysis makes this manuscript interesting and unique. This manuscript will provide a valuable atlas of human vestibular single cell transcriptomics that can be used for future mechanistic and therapeutic

applications.

However, the major conclusions are not novel but as expected.

1. The main criticism is that the designation of these “hair cell precursors” in schwannoma utricles is only based on trajectory analysis (RNA velocity) with additional hybrid gene expression patterns between hair cells and supporting cells plus some immature hair bundle morphology. These are not conclusive. While RNA velocity analysis suggests that they are not dedifferentiating hair cells upon damage, additional methods need to prove this major conclusion. Due to difficulties in human sample collections and manipulation, they could state less confirmative on their conclusion. What about the nuclear migration and other transdifferentiating features of these transition cells?

2. Additional vestibular samples from donors with recent history of antibiotic or cisplatin uses can be more useful than schwannoma patients for providing insights to hair cell regeneration. These could be a new direction for future studies. The authors should explain the nature of stimuli in these schwannoma samples that led to hair cell regeneration. They did mention a hypothesis that schwannoma exert ongoing damage to hair cells.

3. Only two schwannoma and one organ donor utricles were used for scRNA seq analysis. Why not collect more to increase the reproducibility? Although sufficient numbers of cells from these samples are obtained, it would be more convincing to have more than two patients and one donor.

Reviewer #3 (Remarks to the Author):

The study by Wang and a large number collaborators features the generation of a novel human single cell data set of the utricle. Overall the results are clearly presented. The results of the study provides some evidence for differences in gene expression between mouse and human although not very many of these new candidates are validated. While the study has generated a large amount of useful data on cell types and gene expression in the human utricle, the overall novel contribution is limited. The study confirms that hair cells are lost in patients with vestibular schwannoma and confirms that there is probably some level of regeneration in response to that loss. Multiple possible pathways that may be involved in regeneration are identified, but none are tested. Finally, a concerning issue is the fact that degenerating hair cells are not identified in the single cell data set and that Fig S2C suggests that there are almost no transitional hair cells in the VS data set based on location in the SWNE plot. These results could suggest that many of the “transitional” hair cells are actually dying hair cells and that the profile of hair cell regeneration could actually be a profile of degeneration.

The following specific issues should also be addressed:

Line 137: it is inappropriate to call this an hypothesis as the existing data, as cited, has already demonstrated this.

Line 155: What is the evidence that the three markers listed for hair cell precursors actually mark this population? Are there previous publications demonstrating this?

Line 159: similarly, no references are included for any of the listed markers of any other cell type.

Line 174: a key cell type missing from the analysis is hair cells that are dying. If there are new hair cells being generated, doesn't it stand to reason that there should be hair cells that are dying?

Line 210: is not grammatically correct.

Line 225: given the significant differences in gene expression that are reported here, is it possible that the two clusters do not represent Type I and type II hair cells? were any of the new DE genes validated?

Line 254: how were striolar and extrastriolar supporting cell clusters identified?

Line 273: while the cadaveric utricles may not have come from individuals with a history of auditory or vestibular defects, the age of these cadavers almost certainly means there are hair cell deficits.

Line 294: as pointed out above, degenerating hair cells are noted in the epithelium, why are there none in the single cell data set?

Line 350: is the suggestion that the mechanistic framework for human hair cell regeneration will differ from mouse?

Line 352: this section seems contradictory. In looking at Fig S2C, there appear to be essentially no transitional hair cells in the VS data set. Yet the authors suggest there are more transitional hair cells in these utricles. How does this make sense?

Line 385: why are cells that are positive for POU4F3 or GFI1 but negative for MYO7A precursors rather than degenerating hair cells?

Fig 3F: while ADAM11 seems specific to one type of hair cell, there is clearly KCNH6 in some cells that are also positive for ADAM11.

Fig. S2C: There appear to be hardly any transitional hair cells in the VS sample

Fig S2E:

Fig S2E: This image does not show, convincingly, the CD9 is a supporting cell marker. In fact, based on the labeling it could easily be argued it is a hair cell marker.

Fig S3F and F': what is the proof that these represent striolar and extrastriolar hair cells? If genetic markers from mouse can't be used, shouldn't these need to be validated?

REVIEWER COMMENTS

Reviewer #1 (Remarks to the Author):

This manuscript by Wang et al., describes single cell transcriptome analysis of adult utricle from “healthy” donor and patients with a vestibular schwannoma, using 10X pipeline and Seurat package for analysis. They further make use of vestibular samples from organ donors and patients with vestibular schwannoma to confirm few markers expression and hair cells morphology in situ. They finally use slingshot and RNA velocity for trajectory analysis to suggest the presence of hair cells precursors in their dataset and to draw some molecular trajectories between supporting cells and hair cell types.

One main result is the observation, in their single cell data, of 25 clusters of cells, including 2 types of hair cells (type I and type II, as in mice), and possibly 7 cell types or cell states of supporting cells (not confirmed in vivo). The authors describe molecular similarities with published mouse datasets. They also claim divergence; however, previous murine data were during development; moreover, any difference could easily result from the low number of organ donor for the healthy condition (only one), and from the difference in the quality (depth) of the datasets, whether this is in their human data, or in the published mouse datasets. It is therefore too preliminary to conclude any obvious difference. Moreover, the authors should have performed a full comparative between datasets from healthy and schwannoma samples. See specific comments below as well on data analysis and clustering.

1. We acknowledge this reviewer’s concern for the low number of utricle samples from vestibular schwannoma patients and organ donors. We have now added 3 additional vestibular schwannoma utricles and 1 additional organ donor utricle to our study. Specifically, 2 additional vestibular schwannoma utricles and 1 additional organ donor utricle were used for single cell RNA sequencing data and 1 additional vestibular schwannoma utricle was used for histology to characterize hair cell precursors, supporting cell subtypes, and hair cell subtypes (see below).

Regarding scRNAseq comparative analysis, we previously used the published dataset from McInturff et al., 2018 (Biol Open), which contained a limited number of adult (P100) mouse single cells. We have now used a more complete dataset of adult mouse utricle single cells for comparison. This dataset is from 6-14-week-old mice (from 6 animals) and contains 504 type I hair cells, 399 type II hair cells, and 494 supporting cells. Using this mouse and our human single cell RNA sequencing datasets, we now show venn diagrams to illustrate overlapping and non-overlapping hair cell genes in human and mouse utricle (Fig. S3C-D). We also performed Spearman correlation of expressed genes and found a range of correlations between hair cell subtypes (0.628-0.729) and supporting cells (0.829) from mice and human (Fig. S3E). Based on these results, we conclude that there is a modest correlation between mouse and human utricle genes (added to the results section) and have added to the conclusion section that a potential future direction is to study the extent of correlation/divergence in greater details.

As suggested by the reviewer, we added more samples to our single cell RNA sequencing dataset and we are now able to perform direct comparison of organ donor and vestibular schwannoma cell types (newly added Fig. S5 and Table S9). We focused our analysis on the sensory epithelial cell types and simplified the data into four cell types (type I and II hair cells, hair cell precursors, and supporting cells). Comparative analysis revealed some unique genes to each of the categories. We thank the reviewer for this suggestion. Furthermore, readers can now explore comparative analysis of the dataset using the gEAR portal (Fig. S9) (https://umgear.org//index.html?layout_id=human-utricle-sc-atlas).

Also, previous studies have already demonstrated *in vivo* morphological abnormalities and death of vestibular hair cells of the utricle from human with vestibular schwannoma. Some have also shown that supporting cells can spontaneously regenerate hair cells in human utricle with schwannoma, and that overexpressing specific transcription factors could induce ectopic hair cell formation. The principal claim of the authors of the present study is to demonstrate regeneration *in vivo*, and not *ex vivo*, as previously done. While I acknowledge the interest in studying the molecular regulation of hair cell regeneration, this study does not prove the existence *in vivo* of hair cell precursor cells (their description here are subjective, not scientifically based), and their potential to regenerate hair cells, which would have been the main impact of this study, should the authors had demonstrated it. Moreover, they do not compare datasets to prove any increased regeneration, and do not properly demonstrate it *in situ*.

There are multiple short cuts, limited impact in its current form, and many overstatements that preclude publication in this high-profile journal. I strongly suggest improvement of data analysis (+ adding samples), less focus on repeating previously published data (or condense them), and instead confirm the existence of HC precursors *in situ* (using various, new markers) and their potential (*ex vivo*) of differentiating into hair cells.

2. Thank you for these suggestions. Previous studies examining regeneration using vestibular schwannoma utricles rely on culture paradigms and additional treatment with the ototoxic aminoglycosides. We would like to point out that our study uses tissues freshly collected from humans without culture manipulation as pointed out by this reviewer. We leveraged the damaging effects from vestibular schwannoma itself on the inner ear without administering additional ototoxins, thus our findings offer novel insights and are rather different from previous studies on human hair cell regeneration.

We have now added additional utricles from organ donors and vestibular schwannoma patients for single cell RNA sequencing. In addition to the single cell RNA sequencing analyses, our study has 7 samples from organ donors, 20 from vestibular schwannoma patients, and 4 cadaveric specimens, bringing the total number of samples in this study to 37. Unbiased analysis of the new integrated single cell RNA sequencing dataset still revealed a cluster of hair cell precursors from both cohorts, notably when using a new bootstrapping algorithm to detect cell clusters in an unbiased manner (Fig. S2A', S2B-B''). In addition to the 3 sets of hair cell precursor markers we demonstrated in the initial submission (1) GF11+ / MYO7A-negative / SOX2+; 2) POU4f3+ / MYO7A-negative / SOX2+; 3) ANXA2+ / SYT14+ / MYO7A-low) in organ donor and vestibular schwannoma utricles, we have now added 2 additional markers multiplexed with other markers (GPX2 and ATOH1) (new Fig. 7 and S7). Like SYT14, GPX2 was found to be absent in supporting cells, expressed at low levels in hair cell precursors and high in hair cells. *In situ* hybridization confirmed that *GPX2* mRNA was expressed in hair cell precursors from the vestibular schwannoma utricles. They are elongated, spanning from the basement membrane to the luminal surface of the sensory epithelium, contain basally located nuclei, and resemble hair cell precursors previously described (Hewitt et al., 2023, Springer handbook of Auditory Research, Bucks et al., 2017 eLife). Furthermore, we found that hair cell precursors expressed lower levels of MYO7A protein than hair cells (Fig. 7A). Lastly, we detected *ATOH1* mRNA in GPX2+, MYO7A-low, hair cell precursors (Fig. 7D). We measured the nucleus distance from basement membrane and found that nuclei of hair cell precursors were positioned significantly lower than those of hair cells, but not differently from those of supporting cells (Fig. 7E). Overall, these new data confirmed the presence of hair cell precursors in vestibular schwannoma utricles. These have been added to the results section and discussion sections.

Specific Comments:

Title:

should change “inner ears” to “utricle” or “otolith organs”, as it is not the whole inner ear that has been analyzed.

3. We have changed the title.

Intro:

- line 77-78: yes, previous study has shown regenerative capacity of supporting cells, to form new HCs in utricle of schwannoma patients (Taylor RR et al., 2018, eLIFE).

4. We have added the following to the introduction. “Prior studies examining the regenerative capacity of the human inner ear have been limited to showing hair cell regeneration using aminoglycoside-damaged organotypic cultures of human utricles procured from surgical patients.”

Results:

- line 139: only one donor is highly limiting for single cell transcriptomics, even more when data must be compared with pathological samples. Moreover, from the two donors with vestibular schwannoma, one had vestibular symptoms (uncharacterized?), the other not. This may have large influence on the various characteristics of the tumor, its localization and size, and therefore on the cellular and molecular characteristic of the dissected tissue. How can the author analyze and differentiate that? Also, the average age of the schwannoma donors is about 20 years younger than that of the organ donor (Ctr). Knowing the age dependent cellular and molecular changes in the inner ear epithelia, how can this be taken into consideration in their analysis. With the high number of samples, they process for histology, it is unclear to me why they only analyze 1 healthy donor sample and 2 schwannoma samples for single cell RNAseq.

5. We have added another organ donor utricle and 2 vestibular schwannoma utricles for single cell RNA sequencing (4 VS utricles and 2 OD utricles in total), doubling the sample size of our original dataset. Collecting healthy and live inner ear tissues from humans is extremely challenging and our validated single cell transcriptomic atlas represents one of the largest to date, and the first to contain organ donor tissues. We examined inner ear tissues from many VS and OD utricles (20 VS and 7 OD utricles (Table S1)) because we sought to validate the transcriptomic dataset in multiple samples. Transcriptomic datasets are valuable but requires validation because of potential variability among individual VS patients and organ donors as this reviewer has mentioned. Moreover, we validated the genes and proteins of interest using a combination of *in situ* hybridization and immunohistochemistry. We also analyzed cell morphology, innervation patterns, and location of nuclei. We believe these complementary approaches are important in validating the single cell dataset and hence used many samples. We acknowledge that our data does not rule out potential variability among individual VS patients and organ donors, nor was this our intention.

The overall average ages are 51.0 for VS patients and 44.8 for OD and they are not statistically different ($p=0.45$). To assess the potential confounding effects of patient age and other clinical history (audiologic functions, tumor size, vestibular symptoms), we have calculated Spearman's correlation coefficients between these factors and the number of hair cells, supporting cells, hair cell subtypes, and numbers of Pou4f3+ or Gfi1+ hair cell precursors. We found no statistically significant correlation among any of these factors except for a negative correlation between age and the number of Gfi1+ hair cell precursors (Table S1). We have also added the description of these to the results and discussion sections.

- line 153-159: need references to gene sets. Why TMC1 only observed in cluster 13, not in cluster 17.

6. We have listed references to the gene sets (Table S2-7). The reviewer is correct that we have observed TMC1 to be differentially expressed in the type I hair cell cluster (cluster 5) and not the type II hair cell cluster. We have observed a similar difference in the mouse utricle type I and II hair cells also (Jan et al., 2021, Cell Rep). At present, we do not know the reason for this finding but plan on studying this in more detail in future experiments.

Also, in Jan et al., 2021, Cell Rep, S100A6 is described as a non-HCs epithelial marker gene (in utricle). Notably, it seems necessary here to better justify the identity of cluster 20 as hair cell precursors.

7. This is an excellent point. As with many other clustering algorithms, the Seurat algorithm simultaneously analyzes expression of multiple genes to unbiasedly define each cell cluster. One should expect hair cell precursors to share some but not all genes with the supporting cell and hair cell clusters. As stated above, we have now validated additional markers for hair cell precursors (GPX2, ATOH1, POU4F3, GFI1) *in vivo*. We have further used a separate algorithm to determine the most stable and statistically meaningful number of clusters using a bootstrapping mechanism (*chooseR*, Patterson-Cross et al., 2021 BMC Bioinformatics– Fig. S2B). When examining the entire dataset, we observe that the hair cells appear as one cluster, and the supporting cells as 3 clusters, one of which includes the hair cell precursors (Fig. 1). After subsetting to only keep the sensory epithelial cells, we re-ran the *chooseR* algorithm which generated eight clusters that clearly define type I and type II hair cells, hair cell precursor cells and supporting cells (Fig. S2). One cluster of supporting cells is validated as striolar supporting cells (cluster 4). The results of *chooseR* for the entire dataset are now included in the manuscript (Fig. S2B-B''), and *chooseR* was re-run for the subsetting sensory epithelial cells. The *chooseR* plots for the subsetting data is shown here (but not included in the main manuscript, also see Reviewer #1, point #15).

On the left is a Silhouette distribution plot showing tested cluster resolutions (0.2-2). Each tested resolution has dark horizontal black line indicating the median score with the gray box showing the 95% CI. Each dot represents a cluster at each of the tested resolutions. The red vertical line indicates the

algorithm's chosen resolution, and the blue horizontal line is the decision threshold (Patterson-Cross et al., 2021). The chosen resolution of 0.2 was then used in the Seurat clustering step. On the right is an average co-clustering frequency plot at the chosen resolution of 0.2. This shows that when the data is sub-sampled at 80% and over 100 iterations, each cluster co-clusters with itself with high confidence (red). For more details, we invite the reviewers to see Patterson-Cross et al., 2021 *BMC Bioinformatics*.

- line 161: hair cell and supporting cell clusters are somehow supported with expression of few marker genes. But hair cell precursor cluster is defined arbitrarily. There is no data to sustain this claim, which is important for the rest of the study.

8. See above regarding the numerous markers and morphologic analyses used to characterize hair cell precursors.

- line 162: EPCAM expression is not explained, and not restricted to the selected clusters mentioned. Importantly, one can see in Fig. 1D and E that cluster 17 is not clean, but certainly contaminated, and contain different types of cells, with clear Sox2 and EPCAM co-localizing, and absence in most cells of MYO7A. This suggests the presence of supporting cells within cluster 17. It is important to note also that the use of DoubletFinder is not always sufficient to discard all doublets, or contamination. Moreover, SOX2, a supporting cell marker, is not expressed in clusters 5 and 7, depicted as supporting cells in Fig. S2.

9. For EPCAM expression in our new integrated dataset (after adding additional samples), we have found it to be highly expressed in the hair cells (cluster 7), supporting cells (cluster 1, 5 and 6), roof cells (cluster 9), dark cells (cluster 2) (see violin plots in Fig. 1D). As expected, EPCAM expression is relatively low in immune cells, macrophages, stromal cells, pericytes, and vascular cells. Similarly, Sox2 expression is high among supporting cells, type II hair cells, Schwann cells and immune cells and relatively lower in the other clusters (see violin plots in Fig. 1D).

To address the reviewer's concern for supporting cell genes among the hair cell precursor cluster, we will point out that our unbiased clustering algorithm takes into account multiple differentially expressed genes. Thus, it is not surprising that the hair cell precursor cluster shares expression of some genes with hair cells and supporting cells, albeit at different levels as illustrated by their dynamic expression under pseudotime analysis (Fig. 7 and S7). There are a variety of doublet detection algorithms available. We chose *DoubletFinder* (McGinnis et al 2019, Cell Systems) as it is one of the most robust and widely used tools (cited over 1,200 times). We fully acknowledge that computational methods cannot remove 100% of doublets, which is why we emphasized the use of tissues to validate genes and proteins of interest at single-cell and histologic levels (see above point #1). Finally, because of the limited number of cells per sample inherent to the utricle, we do not expect an unusually high number of doublets as the number of cells loaded for capturing did not saturate the capturing capacity. We further point out that EPCAM expression is lower in hair cells than in supporting cells in other datasets, such as Jan et al. (2022) as shown from gEAR expression plot below.

- line 173: without in situ confirmation, this conclusion cannot stand. Clusters do not equal cell types by default, and identity is not confirmed here, but suggested only based on similarities. See above comments for clusters without apparent identities.

10. We have first used bioinformatic algorithms to identify candidate genes that define clusters representing type I and II hair cells. To clarify this, we have modified the text to now state “Together, this dataset represents the **putative** molecular identities of distinct cell types of the adult human utricle.”

- line 179, there is no data, for now, supporting the identity/existence of putative HC precursors. Moreover, sharing few marker genes with HC clusters is not sufficient to claim such identity.

11. We agree with the reviewer and have only described hair cell precursors as putative here since they were identified computationally.

- line 186: need references to previous reports.

12. We have added these references (McInturff et al., 2018, Jan et al., 2021, Wang et al., 2023).

- line 198: there is no comparative analysis of Ctr vs Schwannoma samples to claim this. And how are these cells clustered individually? This should be done in Fig. 1, irrespective of this claim.

13. We have now separately analyzed all cell clusters in organ donor and vestibular schwannoma tissues and found them all to be present in both cohorts (Fig. 1B'). We performed comparative analysis to compare the expression of type I hair cell, type II hair cell, hair cell precursor and supporting cell genes in organ donor and vestibular schwannoma utricles (Fig. S5, Table S9). Description of these results has been added to the results section. Furthermore, we rephrased this sentence to highlight the fact that the common genes between organ donor and vestibular schwannoma are the result of the integration analysis that was performed. Finally, as seen below (Reviewer #1, point #21), when analyzing organ donor and vestibular schwannoma samples as two separate groups without integration, we continue to observe similar groups of cells.

- line 209, and earlier. Data in tables are presented as fold change; raw data on expression levels of all genes, and in all clusters, should be given first, “fold change” is an analysis.

14. Standard differential gene expression is customarily presented in log₂-fold changes; however, we understand the relevance of counts data for each cluster. Expression data for all genes and cell types is available in gEAR which can easily be accessed at https://umgear.org/index.html?layout_id=human-utricle-sc-atlas. Any gene of interest can be identified here on the dimensional reduction plot (e.g. UMAP or SWNE) and in violin plot format. Furthermore, we have now included two additional tables that list the average expression of all genes per cluster for easier accessibility (Table S2 and 3). Finally, all data is available in NCBI’s Gene Expression Omnibus (GSE207817).

- line 211-223: considering a possible contamination or clustering biases, as mentioned above, such type I vs type II dichotomy should be taken very carefully. Also, previous work referred here (Burns et al. 2015 and McInturff et al., 2018) concerns developing inner ear, therefore comparative analysis is difficult due to stage differences, more than considering mice versus human.

15. We have now added an unbiased clustering method that tests the robustness and stability of each cluster using the *chooseR* algorithm (Patterson-Cross et al. 2021, BMC Bioinformatics). This algorithm uses a sub-sampling iterative bootstrapping method to produce stable clusters in an unbiased manner. The resulting resolution from *chooseR* is ingested by the *Seurat* nearest-neighbor clustering algorithm to produce the final clusters. In the initial data with all cells included, we identified 13 clusters that lumps type I and type II hair cells together. However, when we focus on just the sensory epithelial cells (supporting cells, putative hair cell precursors, and hair cells), we note that there are a total of 8 clusters, including type I and type II hair cells. Our *chooseR* metrics here demonstrate the robustness and reliability of each cluster. We have shown the initial results of *chooseR* for the combined integrated dataset with all cells in Fig S2 and describe the results of the subsetted data. The plots from *chooseR* for the subsetted sensory epithelial cells are shown for the reviewer (Reviewer #1, point #7).

We have also successfully used a similar approach to identify and validate type I and II hair cells in the mouse utricle previously (Jan et al., Cell Reports 2019). Here, we have performed additional experiments to validate type I and II hair cell markers in the human utricle (Reviewer #1, point #17).

Previously, we compared our human dataset to only the P100 utricle cells of McInturff et al. (2018, Biol Open), which represent adult mice, although the number of cells from P100 was limited. We have now obtained a new dataset of the adult (6-10 weeks-old) mouse utricle with significantly more cells for comparison. This dataset from our group is currently under review and the manuscript is deposited in pre-print in Research Square (<https://doi.org/10.21203/rs.3.rs-3190105/v1>).

- line 223: Calb1 is not a type I hair cell markers, but a marker for calyx afferents innervating type I HCs (Stone et al., 2021, J Neurosci + numerous, previous, literature on the subject). The absence of markers do not explain differences between mouse and human, but could find other, various explanations, such as data quality and analysis (also, developing murine or chicken utricles must be different to any adult samples). The right panel in Fig. 3H might also explain variations, as clustering does not seem clean, as suggested in previous comments.

Calb1 is used to label type I hair cells (Prins et al., 2020, Oesterle et al., 2008, Cunningham et al., 2002). The stone et al., 2021 paper did not test anti-calb1 as far as I can tell. Maybe it is also expressed in calyx.

16. We agree with the reviewer and acknowledge the need to systematically compare multiple mouse and human genes expressed, and not focusing only on genes not expressed. In our new analyses, we have compared differentially expressed genes of hair cells and supporting cells from human utricle (this manuscript) and adult mouse utricle (newly collected, (<https://doi.org/10.21203/rs.3.rs-3190105/v1>)). Cross-species comparisons have many limitations that are beyond the scope of this manuscript. However, we have now used a systematic method for analyzing the expression data of human and mouse vestibular epithelia (Tosches et al., 2018, Science). In this method, average gene expression (calculated using *Seurat* AverageExpression function) of each cell cluster is calculated and a pairwise cross-species correlation is performed. This is only done for genes that are differentially expressed by clusters in each species and intersecting between the two species. Next, a Spearman rank correlation is performed to obtain a p-value. This is a summary of the methods – please see details in Supplementary Materials of Tosches et al. (2018). When we performed this analysis, we observe (not surprisingly) that in general, there is a significant correlation among the gene expressions of the cell clusters between mouse and human (Fig. S3E). However, as can be seen from the correlation values, the supporting cells between mouse and human have the strongest correlation (Spearman coefficient of 0.83) followed by type I hair cells (0.73), and finally type II hair cells (0.63). We interpret this as there are moderate degrees of similarity between the expression patterns, however, there are also some differences, as shown by our validated marker Calb1 (Fig. S3B, F-H). We hope to perform a more detailed and systematic multi-species comparison in future studies with additional validation experiments and to determine evolutionary relationships.

Several studies (including Prins et al., 2020, J Comp Physiol A Neurothol Neural Behav Physiol Fig. 1 and Stone et al., 2021, J Neuro Fig. 8) have used anti-calb1 to label striolar type I hair cells and calyces in the adult mouse utricle. We have independently verified this staining pattern (Fig. S3F). As Calb1 mRNA was found to be highly expressed in the type II hair cell cluster in the human utricle, we used the same antibody to label sections of human utricles and validated it to be co-expressed with *KCNH6* mRNA in type II hair cells (Fig. S3B). Thus we conclude that Calb1 marks type I hair cells in the mouse utricle and type II hair cells in the human utricle.

- Fig 3F: Discrepancy with text line 235; SOX2 staining is found both in type I (F') and type II (F''), and is cytoplasmic, while it should be nuclear. Similarly, *KCNH6* is also found in both morphological types in the panels. Specificity in panel 3G is very unclear too.

17. We apologize that the SOX2 staining in the initial submission showed high background. We have now improved our protocol and achieved staining specific to nuclei of type II hair cells and supporting cells (Fig. 3F-I). In addition, we have immunostained for TUJ1 to identify calyx innervation specific to type I hair cells. We now show *KCNH6* expression specific to SOX2+, MYO7A+ hair cells that lack TUJ1+ calyx staining (Fig. 3H-I).

- line 244: too preliminary data to conclude that, as previously mentioned, this could be due to data quality and analysis, and age.

18. See above explanation for determining hair cell subtypes and comparing mouse and human utricles.

- Fig. S3A does not clearly establish CRABP1 as a type I hair cell marker. Not obvious neither for VSIG10L2, unclear which hair cell type it labels in the ISH/immunostaining panel.

19. We have now performed *in situ* hybridization for CRABP1 and concurrently immunostained for TUJ1 and SOX2 and found that CRABP1 mRNA expression is exclusive to type I MYO7A+, SOX2-negative hair cells with Tuj1+ calyx (Fig. S3A). Similarly, we found VSIG10L2 mRNA to be selectively expressed in type I MYO7A+, SOX2-negative hair cells with Tuj1+ calyx. These histologic results validate the bioinformatic data predicting CRABP1 and VSIG10L2 as differentially expressed genes in type I hair cells.

- line 254 and related figures: How striolar versus extrastriolar supporting cells were identified unbiasedly is not explained. The striolar supporting cell population represents a very small number of cells that do not seem to separate on their graph (Fig. S3D) from the other supporting cells. Extrastriolar cells in S3.F' do not seem to statistically represent a molecularly different group of cells, with no clear marker genes differentiating them from striolar cells. An explanation for this could be that the striolar cells are also found in the extrastriolar SC group in their dataset. If striolar SC would be unbiasedly identified, one would expect to see them relatively separated from (or at the edge of) the extra SC large cluster in S3D, and not surrounded by it. There are no data to substantiate their conclusion on divergence between mouse and human.

20. This is an excellent point. Using our new scRNAseq dataset that has more samples, we observe that the striolar supporting cells do indeed cluster as a separate cell state (cluster #4) (Fig S2A', Fig 4B). This was done after running *chooseR* to determine the optimal clustering resolution for the sensory epithelial cells. The striolar supporting cells (cluster #4) have their own top differentially expressed genes (Dot plot in Fig. S2E, Table S3), and we have validated, via *in situ* hybridization, three striolar/extrastriolar markers (FRZB, GFAP, SFRP2) (Fig. 4F).

- Fig. 4A-D: marker expression could change (decrease) in schwannoma samples, with no consequence on cell number (density). This is visible in Fig. S4A, B, especially in YZ plan of the B panels. To test this, authors should show at DAPI counter-staining, or similar. Moreover, and in relation to a previous comment, could the authors separate in their scRNAseq analysis and clustering, the schwannoma from Ctr datasets, and analyze differences in expression levels. Can they also then reproduce the clustering of HCs and SCs using the two separated datasets? Anyhow, similar loss of vestibular hair cells and stereociliary abnormalities have been described in past studies in vestibular schwannoma samples (Hizli et al., 2016; Taylor et al., 2015).

21. We have provided DAPI as nuclear staining to confirm that hair cell numbers were significantly lower in the vestibular schwannoma utricles than organ donor and cadaver utricles (Fig. 5E-F). We also added DAPI staining in Fig. S4A'-B''. We have also referenced both the studies this reviewer mentioned since our data builds on their initial findings that vestibular schwannoma leads to vestibular hair cell loss *in vivo*.

While there are a variety of methods for analyzing single cell RNA sequencing data sets from different conditions, we chose to provide an integrated dataset using the Seurat method and believe that integration of the dataset is critical for analysis to reduce batch effect and bring all samples into the same latent spaces. We still sought to test the valid hypothesis posed by this reviewer. We therefore repeated the analysis by looking at vestibular schwannoma utricles and organ donor utricles as two separate groups. Filtered *CellRanger* expression matrices underwent the same analysis detailed in our

methods section (cell/gene level QC, keeping cells with <20% mitochondrial genes, doublet detection, *Seurat* clustering, subsetting data, and differential gene expression). Data (below) shows that type I and type II hair cells and hair cell precursors are present in both the OD and VS samples when analyzed separately.

Organ Donor Analysis

Vestibular Schwannoma Analysis

- line 291: data are not showing preferential loss of type II hair cells, both cell types are greatly affected, with indeed more for type II. This difference could also be accounted by the difficulty in counting HCs in

the schwannoma samples using HC markers. Indeed, the authors use a HC marker (SOX2) to count type II cells (SOX2 levels seem to decrease in a large number of cells in S4B''), while they are using a marker of calyx nerve endings for counting type I cells.

22. We have modified the results section to reflect that both type I and II hair cells have significantly degenerated. We have provided Tuj1 staining to label the type I hair cell calyces to confirm that there are more type I hair cells compared to type II hair cells in VS utricles (Fig. 5G-F).

- line 292: similar to observations previously published (Taylor et al., 2015).

23. Revised.

- line 309: not only suggested but analyzed ex-vivo (Taylor et al., 2015).

24. Revised.

- line 339: this conclusion of a supporting cell-to-hair cell transition is subjective and should be tuned down. Using slingshot as a first method is incorrect since it is biased, the authors choose the trajectory, which however is the first question, and should not be deduced subjectively. Therefore, the authors should perform first velocity and associated package from the S. Linnarsson's lab, for unbiased trajectory inference; I would suggest using two tools as the first approach, they work synergistically, scVelo and CellRank. Then, this can be followed with Palantir or scFATE (from Kharchenko's lab) for actual analysis of the trajectories. Eventually, they can confirm with Slingshot, but other tools are better suited for unbiased analysis.

25. The reviewer is correct that slingshot relies on a manually designated trajectory. Per the reviewer's suggestion, we have employed the CellRank algorithm (v2.0) and used the pseudotime kernel-based approach to calculate the cell-cell transition matrix and project the streamed trajectories onto the subsetted SWNE plot. The trajectory pseudotimes were obtained from the Palantir algorithm. We then used Slingshot to perform the trajectory analysis to draw the lineages as shown (Fig. S6), and finally followed by tradeSeq to analyze the gene dynamics and calculate the generalized additive model for significant genes (Fig. 6F).

Data on Fig. 5B and C are not showing 2 trajectories. The changes in gene expression do not delineate dynamic changes that are directional, which would be expected if there is trajectory of cell types/states. Color coding is not complete, with end points not shown. For type II for instance, it should show trajectory with following clusters order: 7-4-10-(2)-9-20-17, and the color coding.

26. We have modified this figure (now Fig. 6) and present *CellRank* data demonstrating directionality of the hair cell precursors. We further added *Slingshot* and *tradeSeq* analyses for dynamic gene expressions along the predicted trajectories. The cell density plots at the bottom of Fig. 6D and E are now labeled with the cluster numbers that the lineages traverse through. Finally, we have color coded the supporting cells (as one group for ease of visualization), hair cell precursors, and type I/II hair cells in the dynamic expression plots in Fig 6F-F''. We are not sure what the reviewer means by end points not shown.

Data on Fig. 5D are not convincing, as they show selected genes only, yet with a discontinuity in their trajectory of expression levels. Also, to appreciate any trajectory, cells should be labelled in the appropriate color code of the cluster they belong, as in Fig. 5A. Overall, one cannot conclude on an actual differentiation process in vivo. The authors wish to answer whether regeneration occurs in

human in vivo, but such gene expression analysis is artificial in essence and must be proven/assessed experimentally using similar ex vivo methods as in Taylor et al., eLIFE, 2015, onto which they test their molecular pathways.

27. Per reviewer's suggestion, we have now used color coded clusters in pseudotime (new Fig. 6F-F'). In Fig. 6F, we selected dynamically expressed genes based on their patterns of change and also their levels of expression. The reviewer is correct that one cannot make definite conclusion of gene function based solely on this analysis. Rather, this is a predictive model that allows us to identify candidate genes and pathways governing regeneration. For example, we found that GPX2 expression is absent in supporting cells and increased in hair cell precursors and hair cells, whereas GFAP is expressed in both supporting cells and hair cell precursors.

- line 344-346: concluding on the trajectory analysis is incorrect. There is no proven "effects".

28. We have clarified in our writing that this computational analysis models candidate genes driving regeneration.

- line 350: Two things should be done for this conclusion. ONE, that the authors prove, at least ex-vivo, the cellular/molecular trajectory, and TWO, that the data (gene expression) and analysis (see above comments for weaknesses in cluster/computational analysis) are of sufficient quality for identifying molecular trajectories and gene-regulatory networks dynamics in details. Unfortunately, both are not shown.

29. We have clarified in our writing that this bioinformatic analysis helps identify candidate genes driving regeneration.

- line 354-362 and Fig. 6G-H: levels of MYO7A (as mentioned above already) could simply decrease in schwannoma samples. Claiming there are more supposedly HC precursors in schwannoma samples is too preliminary to claim. Also, POU4F3 cells with low levels or absence of MYO7A in schwannoma epithelium in 6H'' are clearly positioned in the HC layer, strongly suggesting these are HCs which has lost MYO7A. It is very well known (and confirmed in this study) that vestibular HCs degenerate in patients with vestibular schwannoma, it is therefore not surprising, and instead expected, to see downregulation of some HC markers in schwannoma samples. In support, while not being a particular focus in the field, decrease/loss of MYO7A has been shown in cochlea hair cells in pathologic conditions (Fei yu et al., Neurotox Res, 2016).

30. The reviewer is correct that hair cells can lose defined markers such as MYO7A during degeneration. However, others have shown that dying hair cells in the mature mouse utricle retain MYO7A expression after aminoglycoside treatment (Monzack et al., Cell Death Diff 2015, Bucks et al., Elife 2017). To demonstrate the identity of hair cell precursors more clearly, we have now used additional molecular markers (GPX2, GFAP, ATOH1). This approach allows us to more clearly demonstrate the elongated shape of these hair cell precursors spanning from the basement membrane to the luminal surface of the sensory epithelium, and that they do not contain pyknotic nuclei. Moreover, we have examined the mitochondrial content, a surrogate marker of cell integrity, and found that hair cell precursors are robust (Review #3, point #1 and Fig. S2E). In addition, we have measured the location of nuclei and found that nuclei of hair cell precursors are significantly lower than those of hair cells, and no different from those of supporting cells (Fig. 7E). Lastly, we have performed CellRank analysis, which is an

independent computational analysis to infer cell trajectories, showing that hair cell precursors are differentiating towards hair cells (Fig. 6A).

- Fig. 6, same comments as for Fig. 5. (see above).

31. We have clarified in our writing that this computational analysis helps identify candidate genes driving regeneration.

- Fig. 6I: definitely needs comparative analysis of scRNAseq data between ctr and schwannoma samples, to show potential absence of cluster 20 in healthy sample. Then comes the issue of the paucity of cell number in cluster 20; only one “healthy” donor is limiting and this could easily explain potential differences, if there would be any. Additional donor samples are necessary.

32. We have now added additional samples from organ donors and vestibular schwannoma patients for single cell analysis. The new analysis still shows the presence of hair cell precursors from both cohort. From a single cell RNA sequencing perspective, it is inappropriate to use the reduced dimensions plot to “count” cells or do compositional analysis. There is a number of technical and methodological limitations for this. This is best explained with an illustrative example in section 17.3 “Why cell-type count data is compositional” of the Single-cell Best Practices book (Heumos, L., et al., 2023 Nat Rev Genet (2023)). We have further summarized this for Reviewer #3, point #12 below. Additionally, the computational data guide our experiments, which makes biologic validation a hallmark of any dataset. We have demonstrated and quantified histological data showing the increased prevalence of hair cell precursors in vestibular schwannoma samples (Fig. 7J-K).

- line 363-388: data are sometimes suggestive (as with POU4F3), sometimes against (as with ATOH1) trajectory from supposedly precursors to HC types. Anyhow, one cannot conclude in vivo trajectory from the analysis; it is overstated to write “To confirm the presence of hair cell precursors in vivo”. line 374: no computational method can validate such trajectory, only experiments can favor or validate, which has not been done.

33. We have clarified in our writing that this bioinformatic analysis helps identify candidate genes driving regeneration.

Reviewer #2 (Remarks to the Author):

Wang et al. describes single cell RNA seq analysis of human vestibula and discovered a variety of cell types and clusters. Some clusters have both similar and divergent gene expression when compared to mouse vestibula. Many of these genes identified are novel and have been confirmed with in situ hybridization again in human vestibular samples. The most interesting findings are the cluster of “hair cell precursors” in schwannoma utricles that resemble regenerating hair cells. These interesting cell populations provided some insight on transitory pathways and new genes/factors that can play roles in regeneration of hair cells in humans. Their results are solid and convincing, and the manuscript is well written and easy to follow. The difficulty in collecting human vestibular samples for analysis makes this manuscript interesting and unique. This manuscript will provide a valuable atlas of human vestibular single cell transcriptomics that can be used for future mechanistic and therapeutic applications. However, the major conclusions are not novel but as expected.

1. The main criticism is that the designation of these “hair cell precursors” in schwannoma utricles is only based on trajectory analysis (RNA velocity) with additional hybrid gene expression patterns

between hair cells and supporting cells plus some immature hair bundle morphology. These are not conclusive. While RNA velocity analysis suggests that they are not dedifferentiating hair cells upon damage, additional methods need to prove this major conclusion. Due to difficulties in human sample collections and manipulation, they could state less confirmative on their conclusion. What about the nuclear migration and other transdifferentiating features of these transition cells?

1. Thank you for this suggestion. We have now modified the language so that our conclusion is less confirmative. This is done in the abstract, results, and discussion. We have also validated additional markers to support our claim that hair cell precursors are indeed present. In addition to 3 sets of markers we provided in the initial submission (GFI1+ / MYO7A-negative / SOX2+; POU4f3+ / MYO7A-negative / SOX2+; ANXA2+ / SYT14+ / MYO7A-negative) to identify hair cell precursors in organ donor and vestibular schwannoma utricles (new Fig. 7 and S7), we now validated 2 additional markers in multiplex fashion. Like SYT14, GPX2 was found to be absent in supporting cells, expressed at low levels in hair cell precursors and high in hair cells in our analysis. *In situ* hybridization confirmed that GPX2 mRNA was expressed in hair cell precursors from the vestibular schwannoma utricles. Furthermore, we found that hair cell precursors expressed lower levels of MYO7A protein than hair cells. Lastly, we detected *ATOH1* mRNA in GPX2+, MYO7A-low, GFAP+ hair cell precursors. Overall, these new data confirmed the presence of hair cell precursors with unique transcriptomes in vestibular schwannoma utricles and support our quantification that more hair cell precursors (GFI1+ / MYO7A-negative / SOX2+; POU4f3+ / MYO7A-negative / SOX2+) reside in the vestibular schwannoma than organ donor utricles.

Per this reviewer's recommendation, we have also characterized the morphology of these putative hair cell precursors. They are elongated, spanning from the basement membrane to the luminal surface of the sensory epithelium, contain nuclei that are near the basement membrane similar to supporting cells, and resemble hair cell precursors previously described (Bucks et al., 2017 eLife). We also measured the nucleus distance from basement membrane and found that nuclei of hair cell precursors were positioned significantly lower than those of hair cells, but not differently from those of supporting cells (Fig. 7E).

2. Additional vestibular samples from donors with recent history of antibiotic or cisplatin uses can be more useful than schwannoma patients for providing insights to hair cell regeneration. These could be a new direction for future studies. The authors should explain the nature of stimuli in these schwannoma samples that led to hair cell regeneration. They did mention a hypothesis that schwannoma exert ongoing damage to hair cells.

2. This an excellent point and we have added this potential future direction to the discussion section. One potential future patient population to study would be Meniere's Disease patients who have failed intratympanic gentamicin and undergo labyrinthectomy. While these are even rarer than vestibular schwannoma patient, it would provide valuable insights as mentioned by this reviewer.

3. Only two schwannoma and one organ donor utricles were used for scRNA seq analysis. Why not collect more to increase the reproducibility? Although sufficient numbers of cells from these samples are obtained, it would be more convincing to have more than two patients and one donor.

3. We agree with this reviewer and have now added 2 additional vestibular schwannoma utricles and 1 additional organ donor utricule for single cell RNA sequencing data. The new integrated dataset yielded similar set of differentially expressed genes for hair cells, supporting cells, and hair cell precursors as

before. In total, we have used 37 human samples in the study (that includes histological analyses). In future studies, we hope to collect even a greater number of samples.

Reviewer #3 (Remarks to the Author):

The study by Wang and a large number collaborators features the generation of a novel human single cell data set of the utricle. Overall the results are clearly presented. The results of the study provides some evidence for differences in gene expression between mouse and human although not very many of these new candidates are validated. While the study has generated a large amount of useful data on cell types and gene expression in the human utricle, the overall novel contribution is limited. The study confirms that hair cells are lost in patients with vestibular schwannoma and confirms that there is probably some level of regeneration in response to that loss. Multiple possible pathways that may be involved in regeneration are identified, but none are tested. Finally, a concerning issue is the fact that degenerating hair cells are not identified in the single cell data set and that Fig S2C suggests that there are almost no transitional hair cells in the VS data set based on location in the SWNE plot. These results could suggest that many of the “transitional” hair cells are actually dying hair cells and that the profile of hair cell regeneration could actually be a profile of degeneration.

1. We acknowledge this reviewer’s concern for mistaking dying hair cells as hair cell precursors. Since we have observed dying hair cells in the vestibular schwannoma utricle, we have taken the following steps to ensure that hair cell precursors are healthy and present and distinct from dying hair cells. First, we have added 2 additional vestibular schwannoma utricles and 1 organ donor utricle for single cell RNA sequencing. This approach allowed us to sample more hair cell precursors particularly from the vestibular schwannoma utricles and demonstrate that hair cell precursors are well represented (154 cells total) (Fig. S2). Second, we have examined and found no pyknotic nuclei among hair cell precursors identified by 5 separate markers (POU4F3, GF11, GPX2, ATOH1, GFAP and ANXA2). Third, we have used CellRank analysis as an independent computational method that examines cell state transitions using a Markov model to predict the directionality of cell differentiation, and found that hair cell precursors have trajectories pointing toward hair cells arguing against de-differentiation (Fig. 6A). Finally, one of the hallmarks of dying cells in single cell RNA-seq datasets is a high percentage of mitochondrial genes. We have filtered out cells expressing more than 20% mitochondrial genes. Furthermore, within the remaining cells that are included in our new SWNE plots, we observe that the percentage of mitochondrial genes is on the lower end in the hair cell precursor population compared to hair cells or some other supporting cells (Fig. S2E).

The following specific issues should also be addressed:

Line 137: it is inappropriate to call this a hypothesis as the existing data, as cited, has already demonstrated this.

2. We have changed this sentence.

Line 155: What is the evidence that the three markers listed for hair cell precursors actually mark this population? Are there previous publications demonstrating this?

3. We have tested several methods to identify hair cell precursors based on the pattern of dynamic gene expression predicted by pseudotime analysis. This includes two additional marker genes of hair cell precursors (Fig. 7A and D).

Line 159: similarly, no references are included for any of the listed markers of any other cell type.

4. We have added these references.

Line 174: a key cell type missing from the analysis is hair cells that are dying. If there are new hair cells being generated, doesn't it stand to reason that there should be hair cells that are dying?

5. Widely accepted single cell RNAseq analyses exclude cells with high-mitochondrial counts that are thought to be dying or compromised cells without distinguishing whether this occurs because of the biology or during the cell dissociation/cell preparation steps. We used a similar analysis pipeline with a somewhat liberal criteria for keeping all cells with 20% or lower of mitochondrial genes (Fig. S1 and S2). When looking at percentage of mitochondrial genes in the remaining dataset, we do see that hair cells have a higher percentage than hair cell precursor cells (Fig. S2E clusters 5 and 6). This may be because hair cells are known to contain higher number of mitochondria given their high metabolic needs (McQuate et al., eLife 2023). The problem with adding back the "dying" cells is that it introduces a great deal of uncertainty rendering the subsequent downstream single cell RNAseq computational analyses difficult to interpret. This will introduce cells and droplets with ambient RNA (i.e. RNA outside of the cell that was captured and sequenced), leading to even more noise in the dataset. Indeed we attempted this technique by not using mitochondrial percentage as a surrogate of cell health, but these droplets/barcodes continue to be filtered out as they do not make the threshold cutoffs for minimum number of genes needed or during the doublet detection steps.

Line 210: is not grammatically correct.

6. We have changed this sentence.

Line 225: given the significant differences in gene expression that are reported here, is it possible that the two clusters do not represent Type I and type II hair cells? were any of the new DE genes validated?

7. We have changed this sentence.

Line 254: how were striolar and extrastriolar supporting cell clusters identified?

8. We have updated our cell clustering method to identify number of cell clusters in an unbiased method (Fig. S2). This new unbiased clustering resolution identified by the *chooseR* algorithm divides the supporting cells into 5 cell groups. One of these groups (cluster 4) uniquely expressed some known

mouse striolar supporting cell markers such as *SFRP2*. We therefore compared this group (cluster 4) against the remaining supporting cell groups (new Fig. 4B-C). By comparing the two supporting cell subtypes, differentially expressed genes were identified (new Fig. 4C), of which we validated *SFRP2* to be enriched in striolar supporting cells, and GFAP and *FRZB* in extrastriolar supporting cells (new Fig. 4F).

Line 273: while the cadaveric utricles may not have come from individuals with a history of auditory or vestibular defects, the age of these cadavers almost certainly means there are hair cell deficits.

9. The reviewer is correct that age can play a role in vestibular hair cell loss, however, our study was not designed to study this effect. We would need higher number of specimens across a wider age range to answer this question that is beyond the scope of this manuscript. The average age of the organ donors in this study is 44.8 years and vestibular schwannoma patients is 51.0 years and these were not significantly different. We have also performed correlation analysis and found no correlation between age, tumor size, hearing functions and the numbers of hair cells, supporting cells, or hair cell subtypes (Table S1).

Line 294: as pointed out above, degenerating hair cells are noted in the epithelium, why are there none in the single cell data set?

10. In order to have robust and reliable single cell RNAseq data, dying cells are excluded, which is a widely accepted practice in single cell RNAseq analysis. This is done at two stages: first, dying cells produce ambient RNA (free-floating RNA) that gets captured in droplets. These droplets may have low number of UMIs (i.e. low number of genes expressed) and will therefore be filtered out during the cell calling steps of the *Cell Ranger* pipeline where the algorithm determines what is a cell versus what is an empty or non-informative droplet. Following this, the initial steps of analysis involve removing cells with low gene counts (for example any cell that has less than 200 genes gets excluded). Next, all cells expressing a high percentage of mitochondrial genes are excluded. In the single cell RNAseq literature, the cutoff thresholds typically used is between 5-20%. We set our threshold at 20% which is on the higher end in order to keep most of our cells including hair cells. As seen in our percentage of mitochondrial genes plot (Fig. S1 and S2), if we lowered this threshold further, we would lose many more cells. Adding back all of the low-quality cells that would include dying/degenerating cells will compromise the downstream computational analyses. We attempted this approach, but the droplets with high mitochondrial percentage continue to get filtered out during the other quality control steps.

Line 350: is the suggestion that the mechanistic framework for human hair cell regeneration will differ from mouse?

11. This is an excellent question and we do not know the answer. We do suggest that one should not assume the mechanism is the same and have stated such in the discussion.

Line 352: this section seems contradictory. In looking at Fig S2C, there appear to be essentially no transitional hair cells in the VS data set. Yet the authors suggest there are more transitional hair cells in these utricles. How does this make sense?

12. Fig. S2C represents the number of cells collected and analyzed, which may not represent the true number of hair cell precursors *in vivo*. For example, it is possible that certain cell types are more difficult to isolate from the vestibular schwannoma. We cannot use the scRNAseq data to “count cells” per se.

There are a number of technical and methodological limitations for this. This problem is best explained with an illustrative example in section 17.3 “Why cell-type count data is compositional” of the Single-cell Best Practices book (Heumos, L., et al., 2023 Nat Rev Genet (2023)). We have summarized the first component of this explanation here:

First, there are too few experimental replicates in single cell RNAseq datasets such as this one leading to large confidence intervals when doing differential abundance analysis. Second, single cell RNAseq is inherently limited in the number of cells captured in each run. All cells within the organ are not captured and sequenced, therefore it is a small sampling of the cells. Therefore, the total number of cells is completely proportional and is a scaling factor. Thus, it is known as compositional data (Aitchison 1982 Journal of the Royal Statistical Society. Series B (Methodological)), meaning relative abundance of all datapoints (cells in this case) adding up to 1. The total adding up to 1 creates a negative correlation between cell type abundances.

To this end, quantification within the tissue using histology is a more accurate way to determine if a cell type is more or less abundant within a particular group, in this case between vestibular schwannoma patients and organ donors (Fig. 7J-K).

Line 385: what are cells that are positive for POU4F3 or GF11 but negative for MYO7A precursors rather than degenerating hair cells?

13. By using additional markers to characterize these putative hair cell precursors, we now show that they are morphologically unique. *CellRank* computational analysis also predicts that they are differentiating towards a hair cell fate. Lastly, mitochondrial content among hair cell precursors was low and suggests that they are robust and not degenerating cells (Fig. S2E).

Fig 3F: while ADAM11 seems specific to one type of hair cell, there is clearly KCNH6 in some cells that are also positive for ADAM11.

14. We performed additional experiments using SOX2 and TUJ1 labeling. It shows that *ADAM11*+ cells are SOX2 negative and innervating by TUJ1+ calyx. On the other hand, *KCNH6* is expressed in SOX2+ hair cells that lack a calyx. Thus, *ADAM11* is a specific marker for type I hair cells and *KCNH6* for type II hair cells (Fig. 3F-I).

Fig. S2C: There appear to be hardly any transitional hair cells in the VS sample

15. Please see point # 12. Fig. S2C represents the number of cells collected and analyzed, which may not represent the true number of hair cell precursors *in vivo*. For example, it is possible that certain types are more difficult to isolate from the vestibular schwannoma (Heumos, L., et al., 2023 Nat Rev Genet (2023)).

Fig S2E: This image does not show, convincingly, the CD9 is a supporting cell marker. In fact, based on the labeling it could easily be argued it is a hair cell marker.

16. We have performed additional experiments to co-stain *CD9* with GFAP. *CD9* staining is mainly in the apical portion of GFAP+ supporting cells (Fig. S2F). On the other hand, hair cell cytoplasm, which is located more basally around their nucleus, shows relatively less labeling.

Fig S3F and F': what is the proof that these represent striolar and extrastriolar hair cells? If genetic markers from mouse can't be used, shouldn't these need to be validated?

17. We have performed additional experiments to validate markers of extrastriolar and striolar supporting cells (new Fig. 4). Single cell analysis predicts that *SFRP2* marks striolar supporting cells and *FRZB* extrastriolar supporting cells. GFAP was found to be more highly expressed in extrastriolar than striolar supporting cells. We validated this expression pattern in sections.

REVIEWER COMMENTS

Reviewer #1 (Remarks to the Author):

The revised manuscript has effectively addressed the majority of concerns raised in the previous version. Nevertheless, several issues should be addressed before publication.

Major concerns:

Figure 1 and Analysis:

In line 176, the authors assert that "this dataset represents the molecular identities of distinct cell types of the adult human utricle." While the inclusion of new samples is appreciated and the integration of datasets may enhance clustering robustness, pooling samples from diverse conditions could potentially introduce clusters visible only in VS samples or solely present in OD condition. This might impact the analysis of differential gene expression crucial for future reference in the adult vestibular organ. The authors should show a separate analysis of OD and VS samples in Figure 1. Additionally, a comprehensive comparative analysis between donor and Schwannoma patients' vestibular organs is essential. Figure S5 appears minimal, providing only six genes per cell type.

The figure presented in response to comments, illustrating OD and VS datasets separately, fails to depict type II hair cells in the VS dataset when isolating sensory cells and supporting cells (??). The authors should clarify/correct.

Figure S1, D. Features: There is a discrepancy in ROD1, with The listed value of 5,425 for ROD1 does not align with the graph. The authors should verify and correct if necessary.

Figure S1, D. Mito%: VS4 exhibits a notably high Mito% value, indicating potentially lower sample quality. The authors should either justify the inclusion of VS4 or consider excluding it. As their dataset and analysis will likely serve as a field reference, ensuring the dataset's solidity is crucial.

Reviewer #3 (Remarks to the Author):

The authors have done a nice job of addressing the concerns of the reviewers. The resulting data sets should be very useful for subsequent analysis of hair cell development in human tissue. Just a few comments

Line 264: The result that there is limited overlap in gene expression between type I and type II hair cells between mouse and human requires a deeper analysis. In particular, if possible, it would be worth labeling human tissue with anti-Spp1 to determine if the difference in expression is real rather than a result of differences in sensitivity related to the single cell analysis.

Section on diversity of supporting cell types. This analysis seems very limited. Given the results from the previous section suggested that gene conservation between mouse and human can be limited, it would be more convincing to see some histological validation of some other markers of striolar and extrastriolar supporting cells.

Figure 3: why were Adam11 and Kcnh6, which were not among the top differentially expressed genes according to panels D and E, selected for validation?

Figure 6D and Figure 7B and C, what is depicted in the vertical bars located to the left of the hear maps?

Finally, perhaps I missed it, but is there an indication of where the data will be deposited?

Reviewer #1 (Remarks to the Author):

The revised manuscript has effectively addressed the majority of concerns raised in the previous version. Nevertheless, several issues should be addressed before publication. Thank you for your kind comments.

Major concerns:

Figure 1 and Analysis:

In line 176, the authors assert that "this dataset represents the molecular identities of distinct cell types of the adult human utricle." While the inclusion of new samples is appreciated and the integration of datasets may enhance clustering robustness, pooling samples from diverse conditions could potentially introduce clusters visible only in VS samples or solely present in OD condition. This might impact the analysis of differential gene expression crucial for future reference in the adult vestibular organ. The authors should show a separate analysis of OD and VS samples in Figure 1. Additionally, a comprehensive comparative analysis between donor and Schwannoma patients' vestibular organs is essential. Figure S5 appears minimal, providing only six genes per cell type.

Thank you for your comments. We have now included a comprehensive, non-integrated analysis of OD and VS conditions in Figure S3. This includes the differentially expressed genes as well as top markers in each condition. We have also added a new table listing all the differentially expression genes per condition in Table S2.

Additionally, we have now expanded the integrated OD/VS dataset analysis that systematically compares the major cell types. This is shown as heatmaps along with their respective tables to replace the prior violin plots in Figure S6 and Table S9.

The figure presented in response to comments, illustrating OD and VS datasets separately, fails to depict type II hair cells in the VS dataset when isolating sensory cells and supporting cells (??). The authors should clarify/correct

Thank you for pointing this out. We have corrected this after re-clustering the data. This now is included in Figure S3 showing separate type I and type II hair cell clusters.

Figure S1, D. Features: There is a discrepancy in ROD1, with The listed value of 5,425 for ROD1 does not align with the graph. The authors should verify and correct if necessary.

Thank you pointing this out. The correct number (1,869) has now been added.

Figure S1, D. Mito%: VS4 exhibits a notably high Mito% value, indicating potentially lower sample quality. The authors should either justify the inclusion of VS4 or consider excluding it. As their dataset and analysis will likely serve as a field reference, ensuring the dataset's solidity is crucial.

Thank you for pointing this out. The reviewer is correct that the mito% is higher in VS4

relative to other VS and OD samples. As a quality control, we have deliberately set the criteria of 20% mitochondrial content (described in methods) and have already removed 3 other samples that did not meet these criteria. VS4 still has a vast majority of cells that are below the 20% cut off and therefore contains useful information.

Reviewer #3 (Remarks to the Author):

The authors have done a nice job of addressing the concerns of the reviewers. The resulting data sets should be very useful for subsequent analysis of hair cell development in human tissue. Just a few comments

Line 264: The result that there is limited overlap in gene expression between type I and type II hair cells between mouse and human requires a deeper analysis. In particular, if possible, it would be worth labeling human tissue with anti-Spp1 to determine if the difference in expression is real rather than a result of differences in sensitivity related to the single cell analysis.

We have used anti-SPP1 antibodies on human vestibular schwannoma tissues and the expression in type I hair cells with TUJ1+ calyces is notably more robust compared to type II hair cells (see figure below). By contrast, our single cell analysis showed that expression of SPP1 mRNA was similar between type I and type II hair cells. This discrepancy may be attributed to differences in protein and RNA expression levels, or to differences in their subcellular localization. The reviewer's point is well taken and the text has been adjusted by removing the following "The mouse type I hair cell marker *SPP1* (OSTEOPONTIN) (McInturff et al., 2018) was notably not significantly enriched in our human dataset (Table S5)".

Section on diversity of supporting cell types. This analysis seems very limited. Given the results from the previous section suggested that gene conservation between mouse and human can be limited, it would be more convincing to see some histological validation of some other markers of striolar and extrastriolar supporting cells.

We agree that validating additional supporting cell genes will be useful and believe that this warrants a more in-depth characterization in a future study. We have added this as a limitation of the current study in the discussion.

Figure 3: why were Adam11 and Kcnh6, which were not among the top differentially expressed genes according to panels D and E, selected for validation?

Both *ADAM11* and *KCNH6* mRNA expression are significantly different between type I and type II hair cells. They were both top differentially genes in our original analysis and were therefore validated, but became slightly lower when additional samples were included. Since they remain differentially expressed and were validated, we conclude that they are useful markers to report.

Figure 6D and Figure 7B and C, what is depicted in the vertical bars located to the left of the heat maps?

The different colors on the left side of the heatmaps represent the 5 different patterns of dynamic gene expression observed in type I and II lineages. We have now added this information to the figures and figure legends.

Finally, perhaps I missed it, but is there an indication of where the data will be deposited?

We have uploaded this data to GEO (GSE207817) and also gEAR, which will be publicly accessible (<https://umgear.org/p?!=human-utricle-sc-atlas>)

REVIEWERS' COMMENTS

Reviewer #1 (Remarks to the Author):

The author did a great job in addressing all comments throughout the revision process. They provide a study that will stand as an excellent reference in the field.

Reviewer #3 (Remarks to the Author):

The authors have adequately addressed all of my concerns.